# JanusVLN: Decoupling Semantics and Spatiality with Dual Implicit Memory for Vision-Language Navigation

**Shuang Zeng**[1,3]*, **Dekang Qi**[3], **Xinyuan Chang**[3], **Feng Xiong**[3], **Shichao Xie**[3], **Xiaolong Wu**[3], **Shiyi Liang**[1,2], **Mu Xu**[3], **Xing Wei**[1,2]†

[1]State Key Laboratory of Human-Machine Hybrid Augmented Intelligence, Xi'an Jiaotong University
[2]School of Software Engineering, Xi'an Jiaotong University    [3]Amap, Alibaba Group

{zengshuang, sy_liang2023}@stu.xjtu.edu.cn, weixing@mail.xjtu.edu.cn,
{qidekang.qdk, changxinyuan.cxy, huanlu.wxl, xumu.xm}@alibaba-inc.com,
{xf250971, tenan.xsc}@autonavi.com

## Abstract

Vision-and-Language Navigation (VLN) requires an embodied agent to navigate through unseen environments, guided by natural language instructions and a continuous video stream. Recent advances in VLN have been driven by the powerful semantic understanding of Multimodal Large Language Models (MLLMs). However, these methods typically rely on explicit semantic memory, such as building textual cognitive maps or storing historical visual frames. This type of method suffers from spatial information loss, computational redundancy, and memory bloat, which impede efficient navigation. Inspired by the implicit scene representation in human navigation, analogous to the left brain's semantic understanding and the right brain's spatial cognition, we propose JanusVLN, a novel VLN framework featuring a dual implicit neural memory that models spatial-geometric and visual-semantic memory as separate, compact, and fixed-size neural representations. This framework first extends the MLLM to incorporate 3D prior knowledge from the spatial-geometric encoder, thereby enhancing the spatial reasoning capabilities of models based solely on RGB input. Then, the historical key-value (KV) caches from the spatial-geometric and visual-semantic encoders are constructed into a dual implicit memory. By retaining only the KVs of tokens in the initial and sliding window, redundant computation is avoided, enabling efficient incremental updates. Extensive experiments demonstrate that JanusVLN outperforms over 20 recent methods to achieve SOTA performance. For example, the success rate improves by 10.5-35.5 compared to methods using multiple data types as input and by 3.6-10.8 compared to methods using more RGB training data. This indicates that the proposed dual implicit neural memory, as a novel paradigm, explores promising new directions for future VLN research. Ours project page: https://miv-xjtu.github.io/JanusVLN.github.io/.

## 1 Introduction

Vision-and-Language Navigation (VLN) is a foundational task in embodied AI, requiring an agent to navigate through unseen environments guided by visual inputs and natural language instructions. Recently, capitalizing on the advanced visual perception and semantic understanding capabilities of Multimodal Large Language Models (MLLMs) Lan et al. (2025a;b); Qi et al. (2026); Li et al. (2025b), a new line of research (Zhang et al., 2025b; Yuan et al., 2025; Liang et al., 2025; Shan et al., 2025) has emerged. These approaches leverage vast-scale training data to adapt MLLMs into VLN models, thereby reshaping the future landscape of VLN research.

---

* Work done during the internship at Amap, Alibaba Group.
† Corresponding author.

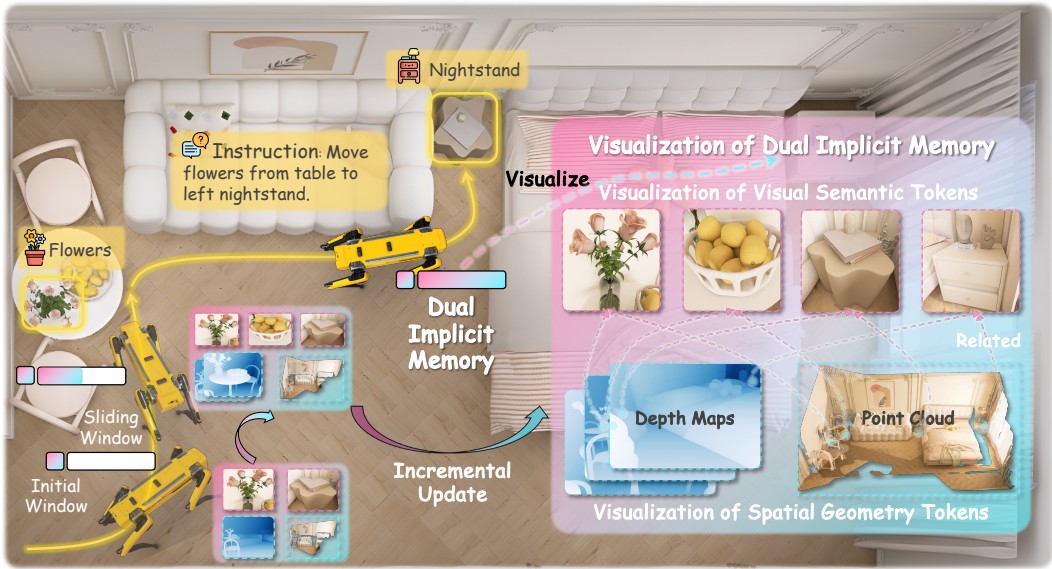

Figure 1: JanusVLN, using RGB-only video, decouples visual semantics and spatial geometry to construct novel, fixed-size dual implicit memory. This memory is incrementally updated during navigation, and its spatial geometry component can be further visualized as depth and point cloud.

To support navigation models in conducting prolonged and effective exploration, these approaches typically only construct an explicit semantic memory. One class of methods (Zhang et al., 2025c; Zeng et al., 2024; Chen et al., 2025c; Liu et al., 2025a) builds a semantic cognitive map using textual descriptions for object nodes and relational edges in the detection method Wang et al. (2026); Qu et al. (2025b;a); Li (2026). However, purely textual descriptions struggle to precisely convey the spatial relationships and orientation of objects, leading to the loss of crucial visual, spatial-geometric, and contextual information. Moreover, repetitive descriptions introduce substantial redundancy and noise. Another class of methods (Cheng et al., 2025; Xiang et al., 2025; Yang et al., 2025c; Li et al., 2025a) stores historical observation frames, which necessitates reprocessing the entire history of observations along with the current frame at each action prediction step, resulting in significant redundant computation. Finally, in both types of approaches, the explicit semantic memory grows exponentially as navigation time increases. This makes it exceedingly difficult for the model to extract critical information from a vast, cluttered, and fragmented memory, thereby leading to severe inefficiency.

More importantly, these methods collectively face a fundamental contradiction. Navigation is an inherently 3D physical interaction, yet the visual encoders of existing VLA models almost exclusively inherit the CLIP paradigm pre-trained on 2D image-text pairs. This approach enables these encoders to excel at capturing high-level semantics while leaving them deficient in understanding 3D geometric structures and spatial information. However, a frequently overlooked yet critical insight is that 2D images are not merely isolated planes of pixels but are projections of the 3D physical world, inherently containing a wealth of 3D spatial cues such as perspective, occlusion, and geometric structures. Whereas human observers can effortlessly perceive depth and comprehend spatial layouts from a single static image, existing models neglect this readily available implicit 3D information in their inputs. This oversight fundamentally constrains their spatial reasoning capabilities in complex navigation tasks.

Inspired by the human brain's hemispheric specialization for navigation, where the left hemisphere handles semantic understanding and the right manages 3D spatial cognition to form implicit representations (Gazzaniga, 1967; Zeng et al., 2025b; Fan et al., 2025b; Li et al., 2025e), we propose a fundamental shift from a single, explicit memory to a dual, implicit neural memory. To this end, we introduce JanusVLN, a dual implicit memory framework for VLN that features both spatial-geometric and visual-semantic memory in Figure 1. We model these two types of memory respectively as fixed-size, compact neural memory, whose size does not grow with the trajectory length. This design is analogous to the human brain's ability to perform efficient memorization within a finite capacity.

To construct this dual implicit memory, we extend the MLLM into a novel VLN model by incorporating a feed-forward 3D visual geometry foundation model, which provides 3D spatial geometric structural information solely from RGB video input, obviating the need for any explicit 3D data. Unlike the visual encoders of general MLLMs, which are predominantly trained on 2D image-text data, this spatial geometry model is typically trained on pixel-3D point cloud pairs, thereby embedding strong 3D perception priors. We establish implicit spatial-geometric and visual-semantic memory by caching historical key-value (KV) from a 3D spatial geometry encoder and MLLM's semantic visual encoder, respectively. These dual implicit memory are dynamically and incrementally updated through the initial and sliding window, enabling the progressive integration of historical information for each new frame without recomputing past frames. Extensive experiments demonstrate that JanusVLN significantly enhances spatial comprehension while lowering inference overhead, achieving SOTA performance on VLN-CE benchmarks. It establishes a new paradigm for VLN research, propelling a shift from being 2D semantics-dominated to 3D spatial-semantic synergy. This marks a pivotal direction toward building the next generation of spatially-aware embodied agents.

In summary, our contributions are as follows:

- We introduce a novel dual implicit memory paradigm for VLN. Inspired by human cognitive science, this framework simultaneously captures visual semantics and spatial geometry to overcome the inherent limitations of existing navigation LLM.

- We unlock the potential of spatial geometric foundation models in streaming VLN. By implementing dual-window and attention fusion mechanisms in VGGT, we efficiently update and integrate historical information incrementally.

- Comprehensive experiments on the VLN-CE benchmark demonstrate that JanusVLN achieves SOTA results without requiring auxiliary 3D data. This validates the efficacy of JanusVLN and establishes a new memory paradigm for the field of VLN.

## 2 RELATED WORK

### 2.1 VISION-LANGUAGE NAVIGATION WITH MULTIPLE VISUAL INPUTS

Vision-Language Navigation (Krantz et al., 2020; Hao et al., 2024; Wu et al., 2025b), the task of guiding an embodied agent to a target location in unseen environments by following instructions, has recently garnered significant attention. Early research (Anderson et al., 2018; Wu et al., 2024; Zheng et al., 2025b; Zhang et al., 2025e) predominantly focused on discrete environments, where an agent navigates by teleporting between predefined nodes. However, these approaches (Hong et al., 2022; Wu & Zhang, 2025; Du et al., 2024; Jin et al., 2026) often exhibit poor performance when deployed on real-world robots operating in continuous 3D spaces. In contrast, more recent studies (Krantz et al., 2020; Wang et al., 2025c; Zhang et al., 2024a; 2023; Zhu et al., 2025b) have concentrated on continuous environments, enabling agents to navigate freely to any collision-free location within simulators. To foster a better spatial understanding and enhance navigational capabilities, some recent works (Wang & Lee, 2025; Xie et al., 2025b; Liu et al., 2023; Wang et al., 2025b; Zhu et al., 2025a) have also begun to investigate monocular RGB-D vision. However, the reliance on additional, expensive hardware for this approach, which is often unavailable in many practical settings, restricts its real-world applicability. In this paper, we propose JanusVLN, a method that enhances spatial understanding using only RGB visual input, eliminating the need for any supplementary 3D data.

### 2.2 MULTI-MODAL LARGE LANGUAGE MODELS FOR RGB ONLY NAVIGATION

The recent, rapid advancement of Multi-modal Large Language Models (Bai et al., 2025; Zeng et al., 2025c; Lin et al., 2025; Ren et al., 2025; Yang et al., 2025d) has injected new momentum the field of Visual Language Navigation. Some approaches (Zhang et al., 2024b; Zhao et al., 2025; Yang et al., 2025a; Fan et al., 2025a; Wei et al., 2025a) have begun to leverage RGB-only video models to build monocular VLN systems, aiming for enhanced generalization and practical value. However, the agents in these studies (Zhang et al., 2025b; Xie et al., 2025a; Cai et al., 2025; Chen et al., 2025b; Qian et al., 2025) typically construct only explicit semantic memory and rely solely on a single, front RGB camera, which poses significant challenges to spatial understanding and often

requires extensive auxiliary data to improve performance. In this paper, we introduce JanusVLN, a VLN framework featuring a dual implicit memory system that encompasses both spatial-geometric memory and visual-semantic memory.

## 2.3 SPATIAL REASONING VIA VISION-LANGUAGE MODELS

Increasing research (Chen et al., 2024a; Zeng et al., 2025; Zheng et al., 2025b; Zhang et al., 2025a; Sun & Li, 2025) efforts have recently aimed to advance the spatial reasoning abilities of Vision-Language Models (VLMs). Previous studies (Chen et al., 2024b; Liu et al., 2025b; He, 2025; Ma et al., 2025; Wei et al., 2025c) have predominantly centered on incorporating 3D data (e.g., point clouds, depth maps) into VLMs to infuse them with explicit spatial information. However, such methods Li et al. (2025d); Hu et al. (2025); Li et al. (2025g); Sun et al. (2019); Wei et al. (2026) often rely on expensive auxiliary hardware, limiting their viability in practical applications. While some recent approaches (Wu et al., 2025a; Zheng et al., 2025a; Jiang et al., 2025; Li et al., 2023; Tan et al., 2025) leverage spatial encoders to derive spatial information directly from videos, they require the entire sequence to be re-processed upon the arrival of each new frame, leading to significant computational redundancy. JanusVLN extracts spatial-geometric features directly from video in an online, streaming fashion. This eliminates repetitive calculations and markedly lowers the inference cost.

## 3 METHOD

### 3.1 PRELIMINARY

**Navigation task definition.** The task of Vision-and-Language Navigation (VLN) in continuous environments is defined as follows. At the timestep $t$, an embodied agent is provided with a natural language instruction $\mathcal{I}$ of $l$ words and an ego-centric RGB video $\mathcal{O}_T = \{x_0, \ldots, x_t\}$, where each frame $x_t \in \mathbb{R}^{3 \times H \times W}$. The agent's goal is to predict a low-level action $a_{t+1} \in \mathcal{A}$ for the subsequent step. The action space is defined as $\mathcal{A} = \{\texttt{Move\_Forward}, \texttt{Turn\_Left}, \texttt{Turn\_Right}, \texttt{Stop}\}$. Each low-level action corresponds to a fine-grained physical change: a small rotation ($30°$), a forward step (25 cm) or stop, which allows for flexible maneuverability in continuous spaces. Upon executing the action $a_{t+1}$, the agent receives a new observation $x_{t+1}$. This process iterates until the agent executes the $\texttt{Stop}$ action at the target location as specified by the instruction.

**Visual geometry grounded transformer (VGGT).** Building upon traditional 3D reconstruction, recent learning-based end-to-end methods (Wang et al., 2025a; Yang et al., 2025b; Liu et al., 2025c; Li & Sun, 2025) employ neural networks to encode scene priors, directly predicting 3D structures from multi-view images. VGGT (Wang et al., 2025a), which is based on a transformer feed-forward architecture, comprises three key components: an encoder for extracting single-image feature, a fusion decoder for cross-frame interaction to generate geometric tokens $G_t \in \mathbb{R}^{\lfloor \frac{H}{p} \rfloor \times \lfloor \frac{W}{p} \rfloor \times C}$, where $p$ is the patch size, and a task-specific prediction head for 3D attributes. The reconstruction pipeline can be formulated as:

$$\{G_t\}_{t=1}^T = \text{Decoder}(\text{Encoder}(\{x_t\}_{t=1}^T)), \quad (P_t, C_t) = \text{Head}(G_t), \tag{1}$$

where a Multi-Layer Perceptron (MLP) head predicts a point map $P_t \in \mathbb{R}^{3 \times H \times W}$ and a per-pixel confidence map $C_t \in \mathbb{R}^{H \times W}$ from these geometric tokens. As our focus is on feature extraction, which embeds 3D geometry prior information, rather than directly outputting 3D attributes, we leverage the encoder and the fusion decoder as our 3D visual geometry encoder.

### 3.2 DUAL IMPLICIT MEMORY

The limitations of traditional explicit semantic memory, including memory inflation, computational redundancy, and the loss of spatial information, coupled with the original VGGT's requirement to reprocess the entire sequence for each new frame, impede the real-time performance and effectiveness of streaming navigation. To address these challenges, we introduce the VGGT as a spatial geometry encoder and propose a novel dual implicit memory paradigm for VLN research in Figure 2. This paradigm models spatial geometry and visual semantics as fixed-size, compact neural

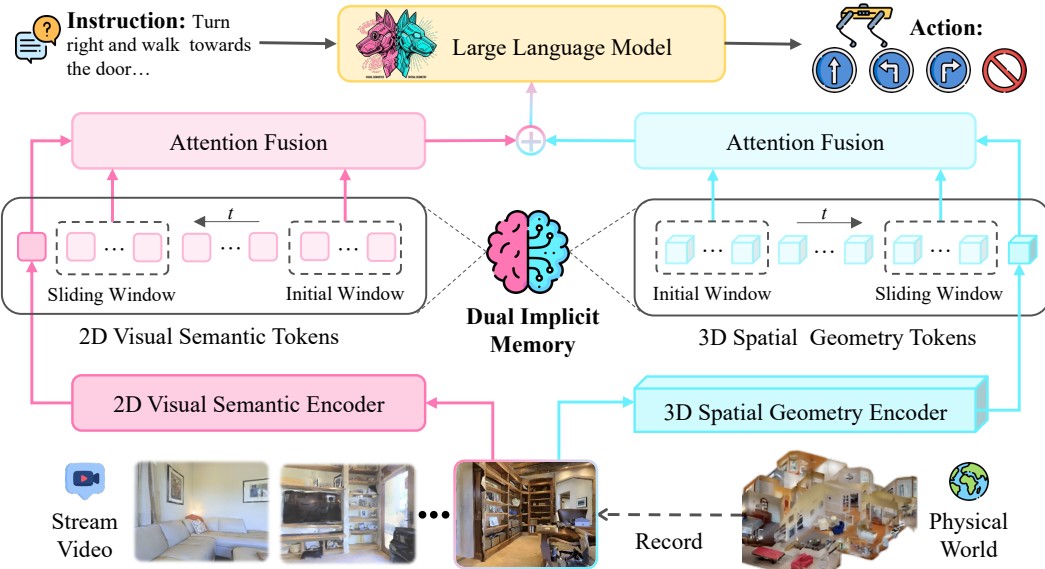

Figure 2: The framework of JanusVLN. Given an RGB-only video stream and navigation instructions, JanusVLN utilizes a dual-encoder to separately extract visual-semantic and spatial-geometric features. It concurrently caches historical key-values from initial and recent sliding window into a dual implicit memory to facilitate feature reuse and prevent redundant computation. Finally, these two complementary features are fused and fed into LLM to predict the next action.

representations by respectively leveraging the history initial and sliding window KV cache of the dual encoders. The spatial memory within the spatial geometry encoder is modeled as follows:

**Implicit neural representation.** In contrast to previous methods that store high-dimensional, unprocessed, and explicit historical frames, we innovatively caches historical KV $M$ that have been deeply processed by neural networks. These KV, derived from the output of attention modules such as transformers, constitute high-level semantic abstractions and structured representations of the past environment. This implicit memory is not merely a compact, efficient storage entity, but a condensed knowledge representation refined by the neural networks. It enables the agent to retrieve and reason over information with minimal computational cost.

**Hybrid incremental update.** For the implicit neural representation, we employ a hybrid cache update strategy instead of caching all historical KV. This approach mitigates the significant memory overhead and performance degradation that arise from extended navigation sequences. The strategy partitions the memory into two components. The first is a sliding window queue $M_{sliding}$ with a capacity of $n$, which stores the KV caches of the most recent $n$ frames in a First-In, First-Out (FIFO) manner. This mechanism ensures the model focuses on the most immediate and relevant contextual information, which is critical for real-time decision-making. When this queue reaches its capacity, the oldest frame's cache is evicted to accommodate the current frame, enabling dynamic incremental updates. The second component permanently retains the KV cache $M_{initial}$ from the initial few frames. The model exhibits sustained high attention weights towards these initial frames, which function as "Attention Sinks" (Xiao et al., 2024; Yang et al., 2026; Lu et al., 2025a; Li et al., 2025f). These sinks provide critical global anchors for the entire navigation task and effectively restore performance. By integrating these two mechanisms, we construct a dynamically updated, fixed-size implicit memory that preserves an acute perception of the recent environment while maintaining a long-term memory of global task information.

For each incoming new frame, we compute cross-attention between its image tokens and the implicit memory to directly retrieve historical information, thereby obviating the need for redundant feature extraction from past frames.

$$G_t = \text{Decoder}(\text{CrossAttn}(\text{Encoder}(x_t), \{M_{initial}, M_{sliding}\})). \tag{2}$$

As shown in Figure 3, VGGT's inference time grows exponentially with each new frame due to its need to reprocess the entire sequence, resulting in an out-of-memory error on 48G GPU with only 48 frames. In contrast, our approach avoids reprocessing historical frames, causing its inference time to increase only marginally and thereby demonstrating excellent efficiency.

For both the semantic encoder and the LLM, namely Qwen2.5-VL, we employ standard KV Cache during the inference stage for acceleration, retaining only the key-value pairs within the initial and sliding windows. Furthermore, these implicit memories and tokens can be visualized to inspect the spatial and semantic information they encapsulate.

### 3.3 JANUSVLN ARCHITECTURE

Building upon the dual implicit memory paradigm, we propose JanusVLN in Figure 2, enhances the spatial understanding capabilities without requiring costly 3D data (e.g., depth).

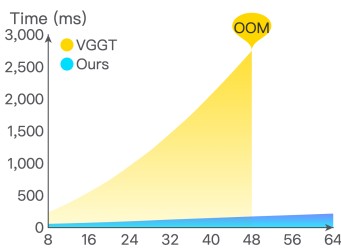

**Decoupling visual perception: semantics and spatiality.** To equip embodied agents with the dual capabilities of semantic understanding ("what it is") and spatial awareness ("where it is and how it's related"), JanusVLN is proposed as a dual-encoder architecture that decouples semantic and spatial information from visual inputs. For 2D semantic encoder, we adopt the original visual encoder from Qwen2.5-VL to interactively encode the input frame $x_t$ with the semantic memory into a semantic tokens:

Figure 3: Inference time comparison for the current frame of varying sequence lengths.

$$S_t = \text{Encoder}_{\text{sem}}(x_t), \quad S_t \in \mathbb{R}^{\lfloor \frac{H}{P} \rfloor \times \lfloor \frac{W}{P} \rfloor \times C}. \tag{3}$$

Additionally, Qwen2.5-VL (Bai et al., 2025) groups spatially adjacent 2×2 patches into a single image token to reduce computational cost, yielding $S'_t \in \mathbb{R}^{\lfloor \frac{H}{2P} \rfloor \times \lfloor \frac{W}{2P} \rfloor \times C}$. For 3D spatial-geometric encoder, we employ the pre-trained encoder and fusion decoder from VGGT (Wang et al., 2025a) model to interactively encode the input frame with spatial memory into spatial-geometric token $G_t$.

**Spatial-aware feature fusion.** Upon acquiring the semantic features $S'_t$ and spatial geometric features $G_t$, we first employ the spatial merging strategy from Qwen2.5-VL (Bai et al., 2025). This strategy concatenates spatially adjacent 2×2 feature blocks within $G_t$ to form $G'_t \in \mathbb{R}^{\lfloor \frac{H}{2P} \rfloor \times \lfloor \frac{W}{2P} \rfloor \times C}$, thereby aligning its shape with that of $S'_t$. Subsequently, we utilize a lightweight two-layer MLP projection layer to fuse the semantic and spatial geometric information:

$$F_t = S'_t + \lambda * MLP(G'_t), \tag{4}$$

where $\lambda$ represents the weight for the spatial geometric features, and $F_t$ denotes the final, spatially-geometrically enhanced visual features. Subsequently, the final visual features, along with the text embedding of instruction $\mathcal{I}$, are fed into the backbone of the MLLM to generate the next action.

## 4 EXPERIMENTS

### 4.1 EXPERIMENTAL SETUP

**Simulation environments and metrics.** Following established methods (Zhang et al., 2025b; Li et al., 2025c; Li, 2025; Chu et al., 2025), we conducted experiments on two of the most recognized VLN-CE (Krantz et al., 2020) benchmark datasets: R2R-CE (Anderson et al., 2018) and RxR-CE (Ku et al., 2020). These datasets comprise trajectories collected from Matterport3D (Chang et al., 2017) scenes using the Habitat simulator (Savva et al., 2019). Consistent with prior work (Cheng et al., 2025; Dai et al., 2025; Yin et al., 2025; Lu et al., 2024), we report performance on the unseen splits using standard VLN metrics, including Navigation Error (NE), Oracle Success Rate (OS), Success Rate (SR), Success-weighted Path Length (SPL), and normalized Dynamic Time Warping (nDTW). Among these, SR and SPL are widely regarded as the primary metrics, reflecting task completion and path efficiency, respectively Wei et al. (2025b); Lu et al. (2025c;b).

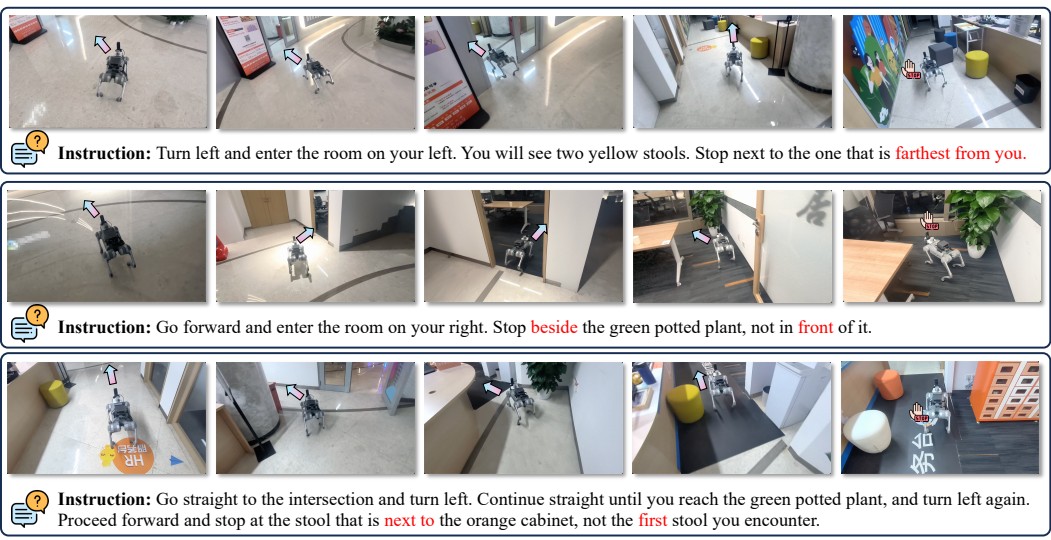

**Instruction:** Turn left and enter the room on your left. You will see two yellow stools. Stop next to the one that is farthest from you.

**Instruction:** Go forward and enter the room on your right. Stop beside the green potted plant, not in front of it.

**Instruction:** Go straight to the intersection and turn left. Continue straight until you reach the green potted plant, and turn left again. Proceed forward and stop at the stool that is next to the orange cabinet, not the first stool you encounter.

Figure 4: Qualitative results of JanusVLN on real-world.

**Real-world evaluation setup.** In real-world experiments, we use the Unitree Go2 as the robotic platform, equipped with an Insta360 X5 camera to capture front RGB. JanusVLN runs on a remote server with an A10 GPU to continuously process RGB and instructions, returning the inference results to the robot for action execution. We focus on navigation tasks requiring spatial understanding.

**Implementation details.** We constructed JanusVLN based on Qwen2.5-VL 7B (Bai et al., 2025) and VGGT Wang et al. (2025a). The model is trained for one epoch, during which we exclusively fine-tune the LLM and the projection layer with learning rates of 2e-5 and 1e-5, respectively, while keeping the semantic and spatial encoders frozen. We set the initial and sliding window size to 8 and 48 frames. The weight for the spatial geometric features $\lambda$ is set to 0.2. For extra data, following StreamVLN (Wei et al., 2025b), we incorporated an additional 155 K trajectories from a subset of the ScaleVLN (Wang et al., 2023c), comprising approximately 9207 K image-action pairs. Furthermore, we employed the DAgger (Ross et al., 2011) algorithm to collect 14 K trajectories (approximately 1485 K image-action pairs) from the standard R2R-CE and RxR-CE datasets.

## 4.2 MAIN RESULTS

**Results on VLN-CE benchmark.** As presented in Table 1 and Table 2, we evaluate our JanusVLN on the two most prominent VLN-CE benchmarks: R2R-CE and RxR-CE. Compared to methods utilizing multiple input types like panoramic views and odometry, JanusVLN achieves a 10.5-35.5 improvement in SR using only a single RGB input, demonstrating the effectiveness of our approach. Furthermore, JanusVLN outperforms SOTA methods that use additional 3D depth data, such as g3D-LF and NaVid-4D, by 12.6-16.7, indicating its ability to effectively enhance spatial understanding with only RGB video streams. Against methods employing explicit textual cognitive maps (e.g., MapNav) or historical frames (e.g., NaVILA, StreamVLN), JanusVLN achieves improvements of 20.8, 10.8, and 3.6, respectively, while using less auxiliary data, highlighting the superiority of its dual implicit memory as a novel paradigm. Furthermore, our method surpasses NaVILA* and StreamVLN* by 10.8-15 in SR when using a comparable amount of data. Notably, even without any additional data, JanusVLN* still outperforms the aforementioned methods that rely on partial extra data by a margin of 3.7-18.8 in SPL. On the RxR-CE dataset, JanusVLN improves the SR metric by 3.3-30.7 over previous methods, demonstrating its superior generalizability. In summary, JanusVLN consistently surpasses various prior methods across all settings, exhibiting strong generalization capabilities. This suggests that the dual implicit memory, as a novel memory paradigm, can effectively replace conventional textual cognitive maps and historical frames.

**Real-world qualitative results.** We selected several navigation tasks that demand spatial understanding in Figure 4, including depth perception (the farthest yellow stool), 3D orientation and rela-

Table 1: Comparison with SOTA methods on VLN-CE R2R Val-Unseen split. External data includes any sources beyond the standard R2R/RxR-CE datasets (e.g., EnvDrop, DAgger, general VQA, etc.). StreamVLN* uses EnvDrop as external data. NaVILA* excludes human-following data. All results are from their respective papers. A training sample is an action or a QA pair. Pano, Odo, Depth, and S.RGB respectively represent panoramic view, odometry, depth, and single RGB.

| Method | Observation | | | | R2R Val-Unseen | | | | Training |
|---|---|---|---|---|---|---|---|---|---|
| | Pano. | Odo. | Depth | S.RGB | NE↓ | OS↑ | SR↑ | SPL↑ | External Data |
| HPN+DN [ICCV21] (Krantz et al., 2021) | ✓ | ✓ | ✓ | | 6.31 | 40.0 | 36.0 | 34.0 | - |
| CMA [CVPR22] (Hong et al., 2022) | ✓ | ✓ | ✓ | | 6.20 | 52.0 | 41.0 | 36.0 | - |
| Sim2Sim [ECCV22] Krantz & Lee (2022) | ✓ | ✓ | ✓ | | 6.07 | 52.0 | 43.0 | 36.0 | - |
| VLN↻BERT [CVPR22] (Hong et al., 2022) | ✓ | ✓ | ✓ | | 5.74 | 53.0 | 44.0 | 39.0 | - |
| Ego$^2$-Map [ICCV23] (Hong et al., 2023) | ✓ | ✓ | ✓ | | 5.54 | 56.0 | 47.0 | 41.0 | - |
| DreamWalker [ICCV23] (Wang et al., 2023a) | ✓ | ✓ | ✓ | | 5.53 | 59.0 | 49.0 | 44.0 | - |
| GridMM [ICCV23] (Wang et al., 2023b) | ✓ | ✓ | ✓ | | 5.11 | 61.0 | 49.0 | 41.0 | - |
| Reborn [ICCV23] (Wang et al., 2023b) | ✓ | ✓ | ✓ | | 5.40 | 57.0 | 50.0 | 46.0 | - |
| InstructNav [CoRL24] (Long et al., 2024) | ✓ | ✓ | ✓ | | 6.89 | - | 31.0 | 24.0 | - |
| COSMO [ICCV25] (Zhang et al., 2025d) | ✓ | | | | - | 56.0 | 47.0 | 40.0 | - |
| AO-Planner [AAAI25] (Chen et al., 2025a) | ✓ | | ✓ | | 5.55 | 59.0 | 47.0 | 33.0 | - |
| LAW [EMNLP21] Raychaudhuri et al. (2021) | | ✓ | ✓ | ✓ | 6.83 | 44.0 | 35.0 | 31.0 | - |
| MapNav [ACL25] (Zhang et al., 2025c) | | ✓ | ✓ | ✓ | 4.93 | 53.0 | 39.7 | 37.2 | - |
| g3D-LF [CVPR25] (Wang & Lee, 2025) | ✓ | ✓ | ✓ | | 5.70 | 59.5 | 47.2 | 34.6 | - |
| Seq2Seq [ECCV20] Krantz et al. (2020) | | ✓ | ✓ | | 7.77 | 37.0 | 25.0 | 22.0 | - |
| NaVid-4D [ICRA25] (Liu et al., 2025b) | | ✓ | ✓ | | 5.99 | 55.7 | 43.8 | 37.1 | - |
| NavMorph [ICCV25] (Xuan Yao & Xu, 2025) | | ✓ | ✓ | | 5.75 | 56.9 | 47.9 | 33.2 | - |
| NaVid [RSS24] (Zhang et al., 2024b) | | | | ✓ | 5.47 | 49.1 | 37.4 | 35.9 | 953K |
| Sim2Real [CoRL24] (Wang et al., 2024) | | | | ✓ | 5.95 | 55.8 | 44.9 | 30.4 | 0K |
| StreamVLN* [arXiv25] Wei et al. (2025b) | | | | ✓ | 6.05 | 53.8 | 45.5 | 41.6 | 10033K |
| Uni-NaVid [RSS25] (Zhang et al., 2025b) | | | | ✓ | 5.58 | 53.3 | 47.0 | 42.7 | 3577K |
| NaVILA* [RSS25] (Cheng et al., 2025) | | | | ✓ | 5.37 | 57.6 | 49.7 | 45.5 | 12574K |
| **JanusVLN* (Ours)** | | | | ✓ | 5.17 | 58.0 | 52.8 | 49.2 | 0K |
| NaVILA [RSS25] (Cheng et al., 2025) | | | | ✓ | 5.22 | 62.5 | 54.0 | 49.0 | 13132K |
| StreamVLN [arXiv25] Wei et al. (2025b) | | | | ✓ | 4.98 | 64.2 | 56.9 | 51.9 | ∼ 26330K |
| **JanusVLN (Ours)** | | | | ✓ | **4.78** | **65.2** | **60.5** | **56.8** | 10692K |

Table 2: Comparison with SOTA methods on VLN-CE RxR Val-Unseen split.

| Method | Observation | | | | RxR Val-Unseen | | | | Training |
|---|---|---|---|---|---|---|---|---|---|
| | Pano. | Odo. | Depth | S.RGB | NE↓ | SR↑ | SPL↑ | nDTW↑ | External Data |
| CMA [CVPR22] (Hong et al., 2022) | ✓ | ✓ | ✓ | | 8.76 | 26.5 | 22.1 | 47.0 | - |
| VLN↻BERT [CVPR22] (Hong et al., 2022) | ✓ | ✓ | ✓ | | 8.98 | 27.0 | 22.6 | 46.7 | - |
| Reborn [ICCV23] (Wang et al., 2023b) | ✓ | ✓ | ✓ | | 5.98 | 48.6 | 42.0 | 63.3 | - |
| AO-Planner [AAAI25] (Chen et al., 2025a) | ✓ | | ✓ | | 7.06 | 43.3 | 30.5 | 50.1 | - |
| LAW [EMNLP21] Raychaudhuri et al. (2021) | | ✓ | ✓ | ✓ | 10.90 | 8.0 | 8.0 | 38.0 | - |
| Seq2Seq [ECCV20] Krantz et al. (2020) | | | ✓ | ✓ | 12.10 | 13.9 | 11.9 | 30.8 | - |
| NavMorph [ICCV25] (Xuan Yao & Xu, 2025) | | | ✓ | ✓ | 8.85 | 30.8 | 22.8 | 44.2 | - |
| Sim2Real [CoRL24] (Wang et al., 2024) | | | | ✓ | 8.79 | 36.7 | 25.5 | 18.1 | 0K |
| Uni-NaVid [RSS25] (Zhang et al., 2025b) | | | | ✓ | 6.24 | 48.7 | 40.9 | - | 3577K |
| NaVILA [RSS25] (Cheng et al., 2025) | | | | ✓ | 6.77 | 49.3 | 44.0 | 58.8 | 13132K |
| **JanusVLN* (Ours)** | | | | ✓ | 6.46 | 51.4 | 44.3 | 59.1 | 0K |
| StreamVLN [arXiv25] Wei et al. (2025b) | | | | ✓ | 6.22 | 52.9 | 46.0 | 61.9 | ∼ 26330K |
| **JanusVLN (Ours)** | | | | ✓ | **6.06** | **56.2** | **47.5** | **62.1** | 10692K |

tive positioning (beside the green potted plant rather than in front of it), and spatial association (the stool beside the orange cabinet). By leveraging the spatial-geometric memory within dual implicit memory, JanusVLN effectively enhances its spatial reasoning, enabling the successful completion of these challenging tasks. For more visualizations, please refer to the supplementary materials.

### 4.3 ABLATION STUDY

In this section, unless otherwise stated, we use no additional data and conduct ablation studies on the R2R-CE benchmark. For more ablation studies, please refer to the supplementary material.

**Ablation of the dual implicit memory.** The ablation study for dual implicit memory is presented in Table 3. Removing the spatial memory led to a substantial drop in the SPL score from 49.2 to 40.9. This finding demonstrates that the spatial-geometric memory effectively enhances the agent's spatial understanding. Furthermore, removing the semantic memory results in a 13.8% decrease in the SR, underscoring the necessity of the semantic memory. Finally, the simultaneous removal of both memory modules leads to a near-collapse in model performance. In summary, these experiments highlight the complementary and indispensable nature of our proposed dual implicit memory.

Table 3: The ablation experiments of each component of the proposed JanusVLN.

| Method | NE↓ | OS↑ | SR↑ | SPL↑ |
|---|---|---|---|---|
| JanusVLN | **5.17** | **58.0** | **52.8** | **49.2** |
| w/o Spatial Implicit Memory | 6.58 | 54.3 | 47.0 | 40.9 |
| w/o Semantic Implicit Memory | 6.75 | 53.1 | 45.5 | 40.0 |
| w/o Dual Implicit Memory | 7.85 | 36.9 | 24.8 | 16.8 |

**Ablation of 3D geometric priors.** We provide an ablation study in Table 4 to investigate the effect of introducing additional encoders. When the spatial geometric encoder VGGT in JanusVLN is replaced by other visual encoders (e.g., DINOv2 (Oquab et al., 2023), and SigLIP 2 (Tschannen et al., 2025)), the performance did not significantly improve. The reason is that these alternative encoders are generally pre-trained on 2D image-text pairs. While this makes them proficient in capturing high-level semantics, this information is largely redundant with that from the original visual encoder of Qwen2.5-VL, and consequently, offers no significant improvement. Conversely, VGGT, being pre-trained on pixel-to-3D point cloud pairs, contributes complementary information. Moreover, a randomly initialized VGGT, devoid of pre-trained 3D spatial-geometric priors, showed no notable gains. This demonstrates that the advantage of JanusVLN lies in its enhanced spatial comprehension, rather than simply increasing model parameters.

Table 4: Comparison between additional, different semantic encoders and spatial encoder.

| Encoder | NE↓ | OS↑ | SR↑ | SPL↑ |
|---|---|---|---|---|
| JanusVLN w/o extra encoder | 6.58 | 54.3 | 47.0 | 40.9 |
| JanusVLN w/ extra DINOv2 | 6.44 | 55.4 | 47.5 | 41.5 |
| JanusVLN w/ extra SigLIP 2 | 6.38 | 55.2 | 47.9 | 41.9 |
| JanusVLN w/ extra VGGT[random init] | 6.61 | 54.7 | 47.2 | 40.8 |
| JanusVLN w/ extra VGGT | **5.17** | **58.0** | **52.8** | **49.2** |

Table 5: Inference time and performance comparison for the current frame of varying sequence lengths between cached memory and VGGT for the online setting.

| Memory Size | Inference Time | NE↓ | OS↑ | SR↑ | SPL↑ |
|---|---|---|---|---|---|
| VGGT (8) | 268 ms | 5.99 | 56.2 | 50.2 | 45.0 |
| VGGT (32) | 1549 ms | 5.66 | 56.8 | 51.2 | 47.6 |
| Cached Memory (8) | 82 ms | 5.91 | 56.0 | 50.5 | 45.7 |
| Cached Memory (32) | 149 ms | 5.52 | 57.1 | 51.7 | 48.3 |
| Cached Memory (48) | 195 ms | **5.17** | **58.0** | **52.8** | 49.2 |
| Cached Memory (64) | 244 ms | 5.27 | 57.5 | 52.3 | **49.4** |
| Cached Memory[w/o initial's KV] (48) | 171 ms | 5.66 | 56.8 | 51.0 | 47.5 |

**Ablation on memory size.** We present the ablation studies on memory size in Table 5. First, as shown in the first row, with a memory of 8 frames, the original VGGT model without caching necessitates re-computation of the entire sequence for each new frame's feature extraction. This results in an inference overhead of 268 ms. Furthermore, as the memory size increases, the inference overhead of VGGT grows exponentially, rendering it impractical for real-world applications. In contrast, our JanusVLN dynamically caches historical KV, eliminating the need for re-computation. This approach significantly reduces inference overhead by 69%-90% while also yielding a slight performance improvement, thereby demonstrating the effectiveness of the implicit neural memory. As the memory size increases, JanusVLN's performance progressively improves, saturating at 48 frames. This suggests that a compact, fixed-size implicit memory is sufficiently effective. Finally, when we omit the preservation of the initial window's KV, a slight performance degradation is observed, indicating that the first few frames of memory do indeed capture significant model attention.

## 5 CONCLUSION

This paper introduces JanusVLN, a novel VLN framework and the first to feature a dual implicit neural memory. Inspired by the implicit scene representation in human navigation, which integrates left-brain semantic understanding with right-brain spatial cognition, JanusVLN constructs two complementary, fixed-size, compact neural memory. This approach overcomes the bottlenecks of traditional methods in memory inflation, computational redundancy, and the absence of spatial perception. By synergistically integrating a MLLM with a feed-forward 3D spatial geometry foundation model, JanusVLN achieves perception of spatial geometric structures solely from RGB video, obviating the need for auxiliary 3D data. The dual implicit memory are derived from the historical KV caches of a spatial geometry encoder and a semantic visual encoder, respectively. They are updated with high efficiency through an incremental process that retains only initial and sliding window of KVs, thus avoiding re-computation. Extensive experiments demonstrate the superiority of JanusVLN, steering VLN research from 2D semantics-dominant toward 3D spatial-semantic synergy, a critical direction for developing next-generation spatial embodied agents.

ACKNOWLEDGMENTS

This work was support by the National Natural Science Foundation of China No. 62572385, the Fundamental Research Funds for the Central Universities No. xxj032023020, and CAAI-CANN Open Fund, developed on OpenI Community.

ETHICAL STATEMENT

We anticipate that JanusVLN technology will advance the application of embodied AI in beneficial domains, such as providing navigational assistance for the visually impaired, improving task efficiency in domestic service robots, and performing search and rescue operations in disaster scenarios. We also recognize that any advanced autonomous navigation technology presents a potential for misuse in negative applications like unauthorized surveillance or military operations, a challenge known as the dual-use problem. The fundamental motivation of this research is to foster scientific progress and social welfare. We condemn any use of this technology for unethical or malicious purposes and call upon the academic community to jointly establish and abide by guidelines for the responsible development and application of AI.

REPEATABILITY

To ensure the reproducibility of our research, the implementation details of JanusVLN are provided in Section 4.1. To foster academic exchange and technical transparency, we will publicly release our source code, model configurations, and fine-tuned model weights in accordance with relevant licenses. This will enable other researchers to replicate our findings and build upon our work.

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

## A   THE USE OF LARGE LANGUAGE MODELS (LLMS)

In this paper, the application of Large Language Models (LLMs) was strictly limited to enhancing writing quality. Upon the completion of the manuscript, we employed Gemini 2.5 Pro (Comanici et al., 2025) to refine the text and identify grammatical or stylistic errors. The model was guided by the following prompt: "You are a top-tier academic expert specializing in refining academic papers. Please polish this text, identify any writing errors, and ensure the original meaning is preserved without altering its substantive content."

## B   MODEL STRUCTURE DETAILS

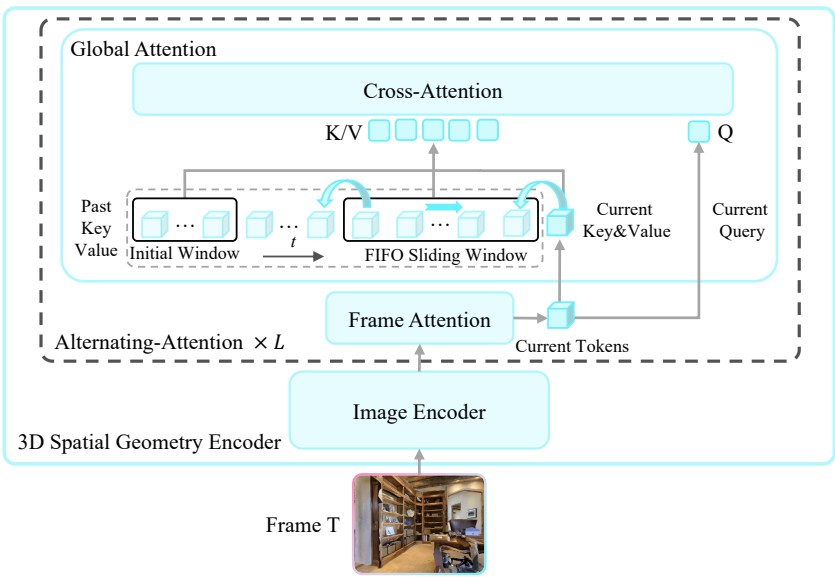

Figure 5: Details of the implicit memory of the spatial geometric encoder.

In the original VGGT, frame attention and global cross-frame attention are executed alternately. In Figure 5 Our spatial encoder, in contrast, fuses information through interaction with a cache during the global attention process. Specifically, the tokens of the current frame first pass through frame attention to establish a local context. Then, during global attention, these current-frame tokens generate the Query. The final Key and Value are constructed by concatenating the historical KV cache with the newly generated KV from the current frame, which are then used to compute the attention. This alternating execution of frame attention and global attention is repeated.

Qwen2.5-VL employs the standard KV Cache mechanism typical of LLMs. Visual embeddings derived from new frame via the semantic encoder generate Queries within the language model. These Queries then compute attention against the Keys and Values of all historical tokens combined with the Keys and Values generated by the tokens of the current frame.

## C   MORE ABLATION STUDIES

**Real world quantitative results.** In our real-world experiments, we employed a Unitree Go2 robotic platform equipped with an Insta360 X5 camera to capture forward-facing RGB images. The JanusVLN model operates on a remote server with an A10 GPU, continuously processing RGB images and instructions, and sending the inference results back to the robot for execution. For quantitative real-world evaluation, we used 25 instructions, each repeated three times, covering both general and spatial understanding tasks. A trial is considered successful if the robot stops within 1 meter of the target. As shown in Figure 6, JanusVLN outperforms its variant without spatial memory across all scenarios. Notably, it achieves a 23.6% improvement on navigation tasks that require spatial understanding, which demonstrates the effectiveness of JanusVLN.

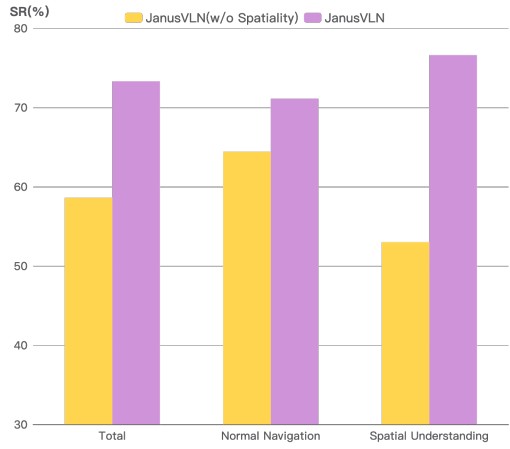

Figure 6: Quantitative experiments in the real world.

Table 6: Comparison on recent HM3D-OVON(Yokoyama et al., 2024b) val unseen.

| Method | SR↑ | SPL↑ |
|---|---|---|
| VLFM [ICRA24] (Yokoyama et al., 2024a) | 35.2 | 19.6 |
| DAgRL+OD [IROS24] (Yokoyama et al., 2024b) | 37.1 | 19.8 |
| Uni-Navid [RSS25] (Zhang et al., 2025b) | 39.5 | 19.8 |
| MTU3D [ICCV25] (Zhu et al., 2025c) | 40.8 | 12.1 |
| JanusVLN | **44.9** | **31.7** |

**Results on recent HM3D-OVON.** As shown in Table 6, we also test on the more diverse, updated HM3D-OVON (Yokoyama et al., 2024b) benchmark. Our approach JanusVLN surpasses SOTA methods by boosting the Success Rate (SR) from 40.8% to 44.9%, which showcases its strong generalization capabilities.

**Ablation of fusion strategies.** Table 7 presents the results for different feature fusion strategies. We varied the weight of spatial features from 0.5 to 0.1 and observed that the performance peaked at 0.2. We also utilize a fusion strategy of Concat and Cross-Attention, where Cross-Attention, despite exhibiting competitive performance, remains marginally inferior to the simple and lightweight addition method. The exploration of more sophisticated strategies is left for future work.

Table 7: Ablation experiments on the fusion strategies of spatial features and semantic features.

| Fusion Strategy | NE↓ | OS↑ | SR↑ | SPL↑ |
|---|---|---|---|---|
| $\lambda = 0.5$ | 5.61 | 55.5 | 50.4 | 46.9 |
| $\lambda = 0.2$ | **5.17** | 58.0 | **52.8** | **49.2** |
| $\lambda = 0.1$ | 5.69 | 55.8 | 50.2 | 46.6 |
| Concat | 5.78 | 55.2 | 49.4 | 45.7 |
| CrossAttn | 5.24 | **58.2** | 52.1 | 48.6 |

**Data Ablation.** Table 8 presents the ablation studies on the use of supplementary data. Notably, even without any additional data, JanusVLN outperforms prior methods that utilized partial supplementary datasets, demonstrating its robust intrinsic navigation capabilities. Following StreamVLN, we observe that incorporating data from ScaleVLN and DAgger individually both yield performance improvements. Furthermore, following StreamVLN, the concurrent use of both data sources leads to further enhancement, showcasing the model's excellent data efficiency. The integration of even

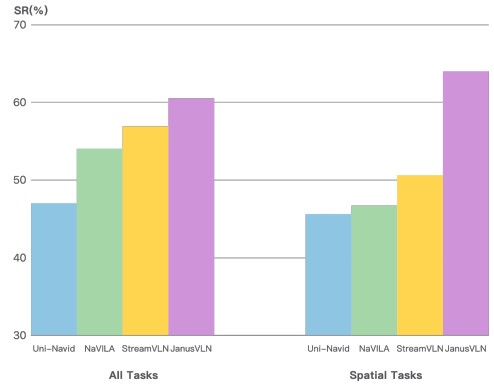

Figure 7: Performance on spatial understanding tasks.

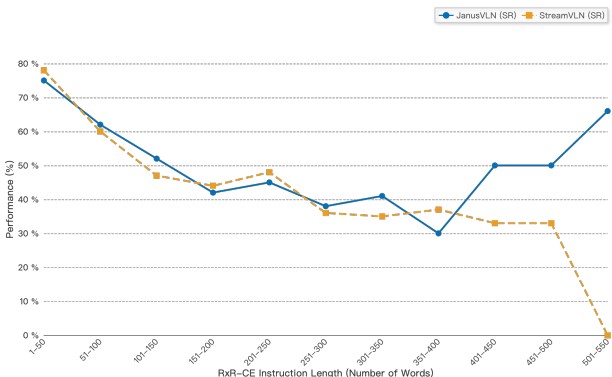

Figure 8: Performance on various instruction lengths/complexity.

larger-scale external datasets, akin to the approaches of StreamVLN and NaVILA, is reserved for future work to construct more powerful navigation agents.

Table 8: Ablation study of different training data compositions.

| Data Compositions | NE↓ | OS↑ | SR↑ | SPL↑ |
|---|---|---|---|---|
| JanusVLN w/o Extra Data | 5.17 | 58.0 | 52.8 | 49.2 |
| JanusVLN w/ ScaleVLN | 5.08 | 62.8 | 55.5 | 50.9 |
| JanusVLN w/ DAgger | 5.02 | 63.4 | 56.4 | 51.7 |
| JanusVLN w/ ScaleVLN & DAgger | **4.78** | **65.2** | **60.5** | **56.6** |

# D STATISTICAL ANALYSIS

**Success and strengths analysis.** In Figure 7, We measured the success rate on instructions requiring spatial understanding (i.e., those containing terms like 'farthest,' 'nearest,' 'larger,' 'smaller,' 'rightmost,' 'leftmost,' 'first,' 'second,' 'front,' 'back,' etc.). We find that the superiority of JanusVLN over prior methods is more pronounced in scenarios requiring spatial understanding than its average gain across all tasks, demonstrating its strong spatial awareness.

**Performance by instruction length.** We analyzed the trends in SR and SPL for both StreamVLN and JanusVLN as instruction length increases in Figure 8. Both models achieve high SR and SPL

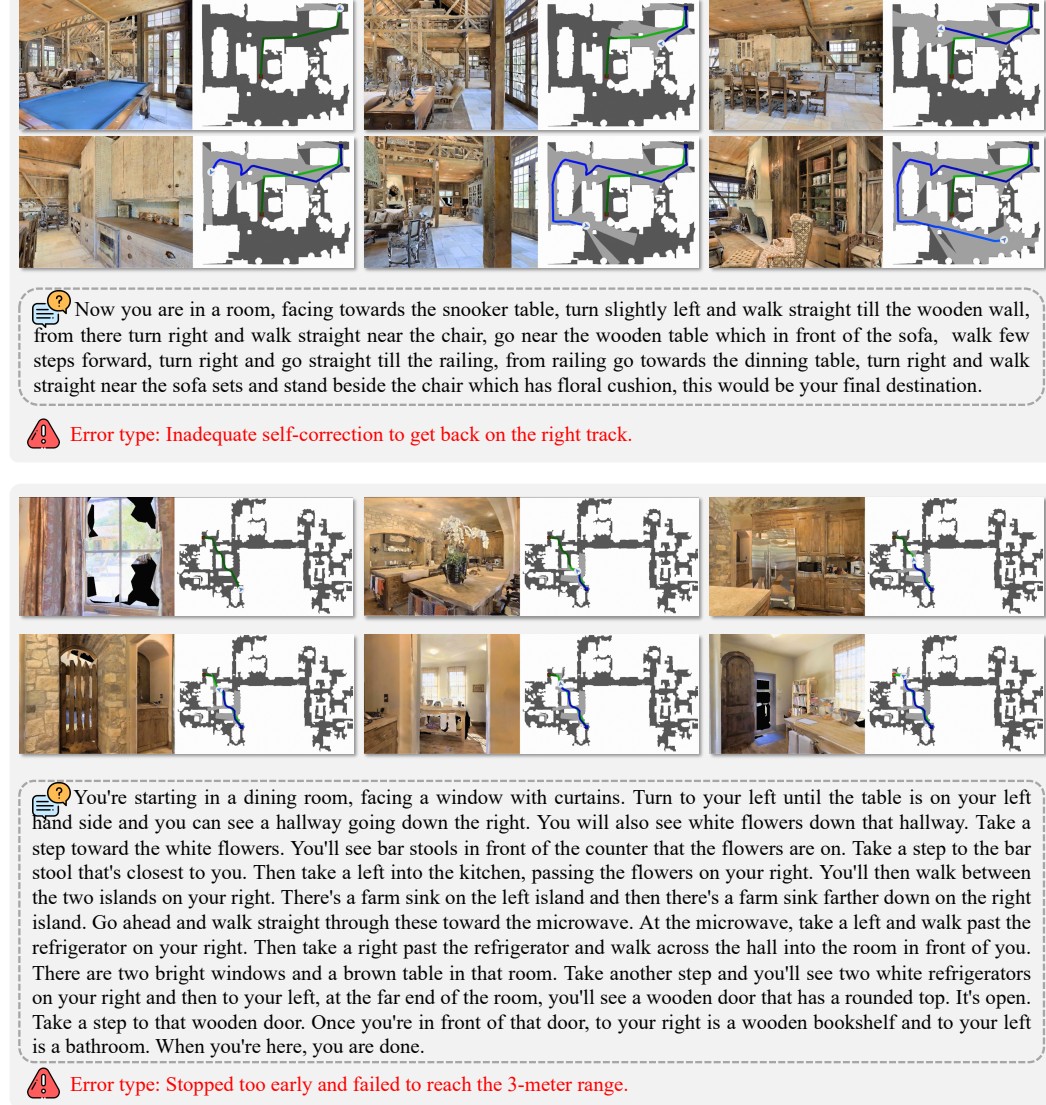

Figure 9: Visualization and presentation of the types of failure cases.

on relatively simple instructions (1-150 words). However, their performance declines on moderately complex instructions (150-400 words), indicating a need to enhance the models' ability to decompose and comprehend complex directives. For the most complex instructions (400-550 words), StreamVLN's performance continues to degrade, eventually reaching zero. In contrast, JanusVLN's performance improves, benefiting from its dual implicit memory paradigm. This is likely because these lengthy instructions provide highly detailed, step-by-step guidance that the model can effectively follow.

**Failure case analysis.** Our statistical analysis reveals two predominant types of failure cases for JansuVLN in Figure 9. First, when the agent deviates from the optimal trajectory, it attempts to correct its course but often fails to recover, leading to compounding errors and eventual failure. Although we collected a limited amount of non-optimal trajectory data via DAgger, it is insufficient to enable robust error correction. Second, JanusVLN appears to employ an overly aggressive stopping policy, sometimes halting prematurely upon sighting the destination and thus failing to enter the success radius. This may be because the spatial information from its VGGT encoder lacks real-world scale, resulting in inaccurate distance estimation.

# E    MORE QUALITATIVE RESULTS

**Visualization analysis of spatial geometric tokens.**    We demonstrate how spatial geometry tokens aid navigation by visualizing them as depth maps and point clouds in Figure 10. In the first example, the depth map derived from the tokens captures precise depth information, enabling a more accurate localization of the farthest chair. In the second, the point cloud constructed from the tokens clearly reveals the chair behind the sink counter. In the third example, both visualizations distinctly represent the size of the door. Finally, in the fourth example, visualizations reveal that the tokens focus on the rightmost house, as reflected in both its depth map and point cloud. In conclusion, the spatial information captured by these tokens is crucial for spatial understanding.

**More qualitative results.**    This section presents further qualitative analysis of JanusVLN in both real-world and simulated environments. For real-world settings in Figure 11, we selected navigation tasks that involve simple and complex instructions, diverse sites, and spatial understanding, where JanusVLN demonstrates excellent generalization. For simulated environments in Figure 12 and 13, we chose complex trajectories and long instructions from the unseen validation sets of R2R-CE and RxR-CE. Leveraging its dual implicit memory, JanusVLN effectively follows these instructions to complete challenging navigation tasks.

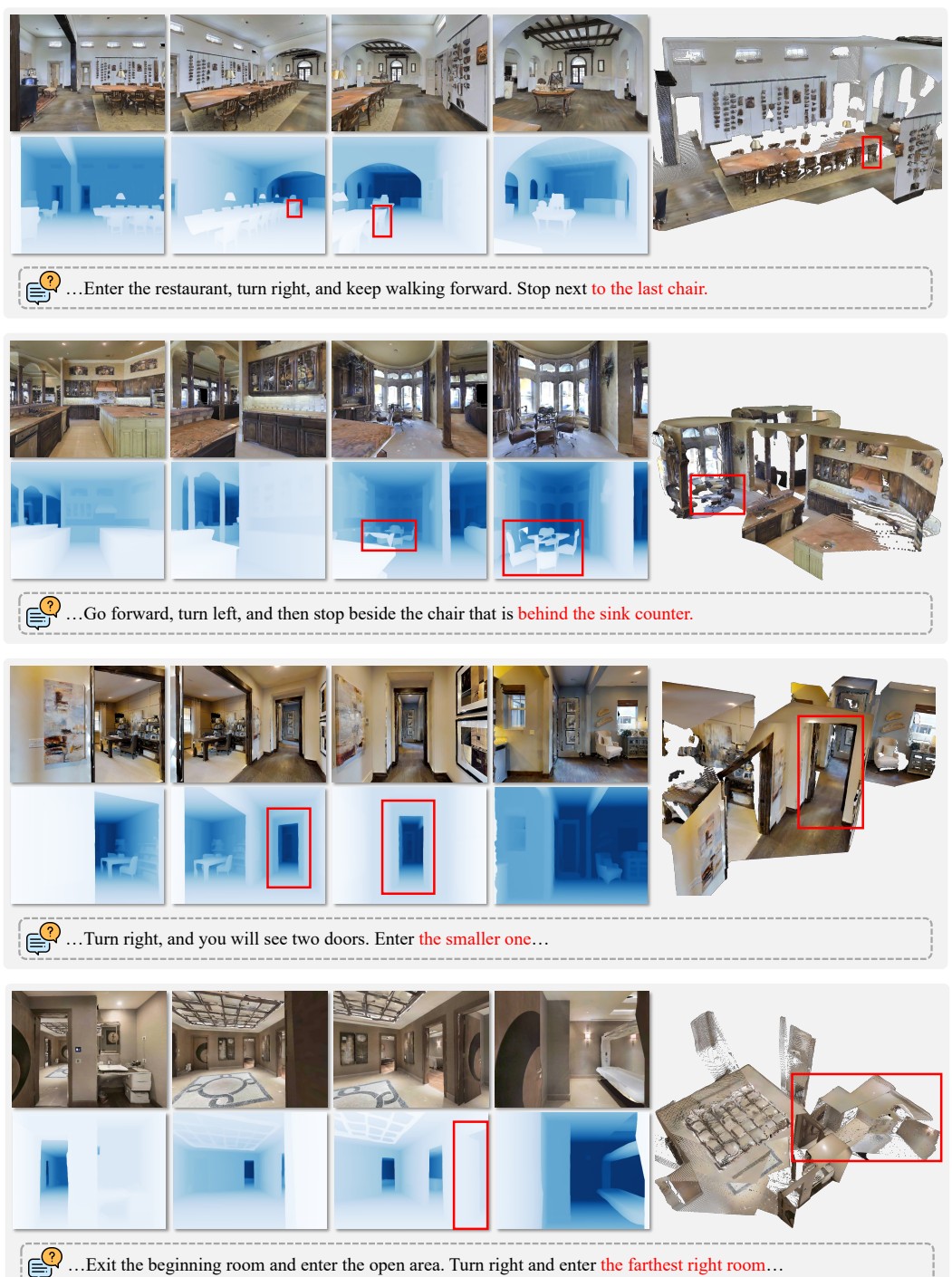

Figure 10: The analysis on the effectiveness of spatial gemetric tokens.

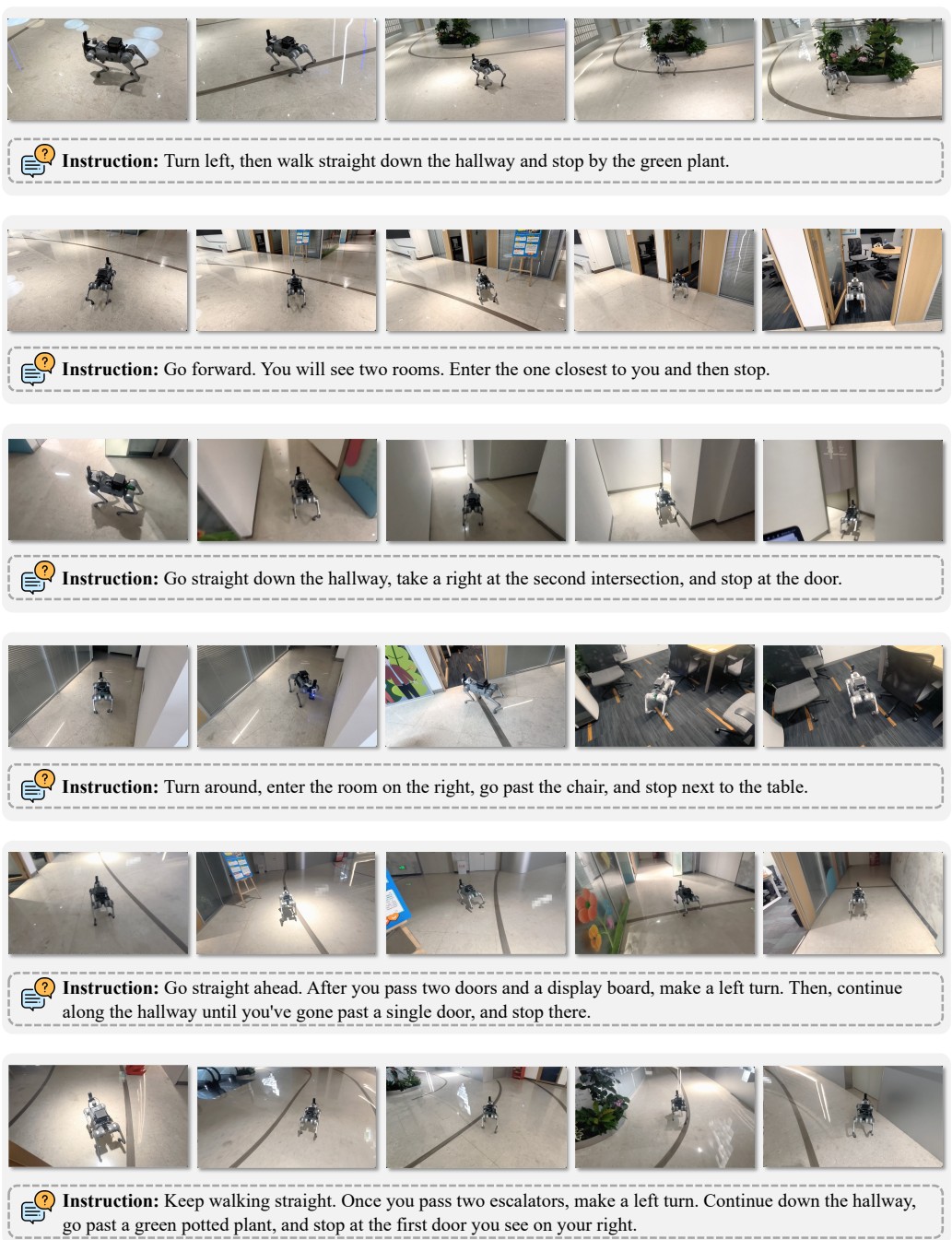

Figure 11: Qualitative results of JanusVLN on real-world.

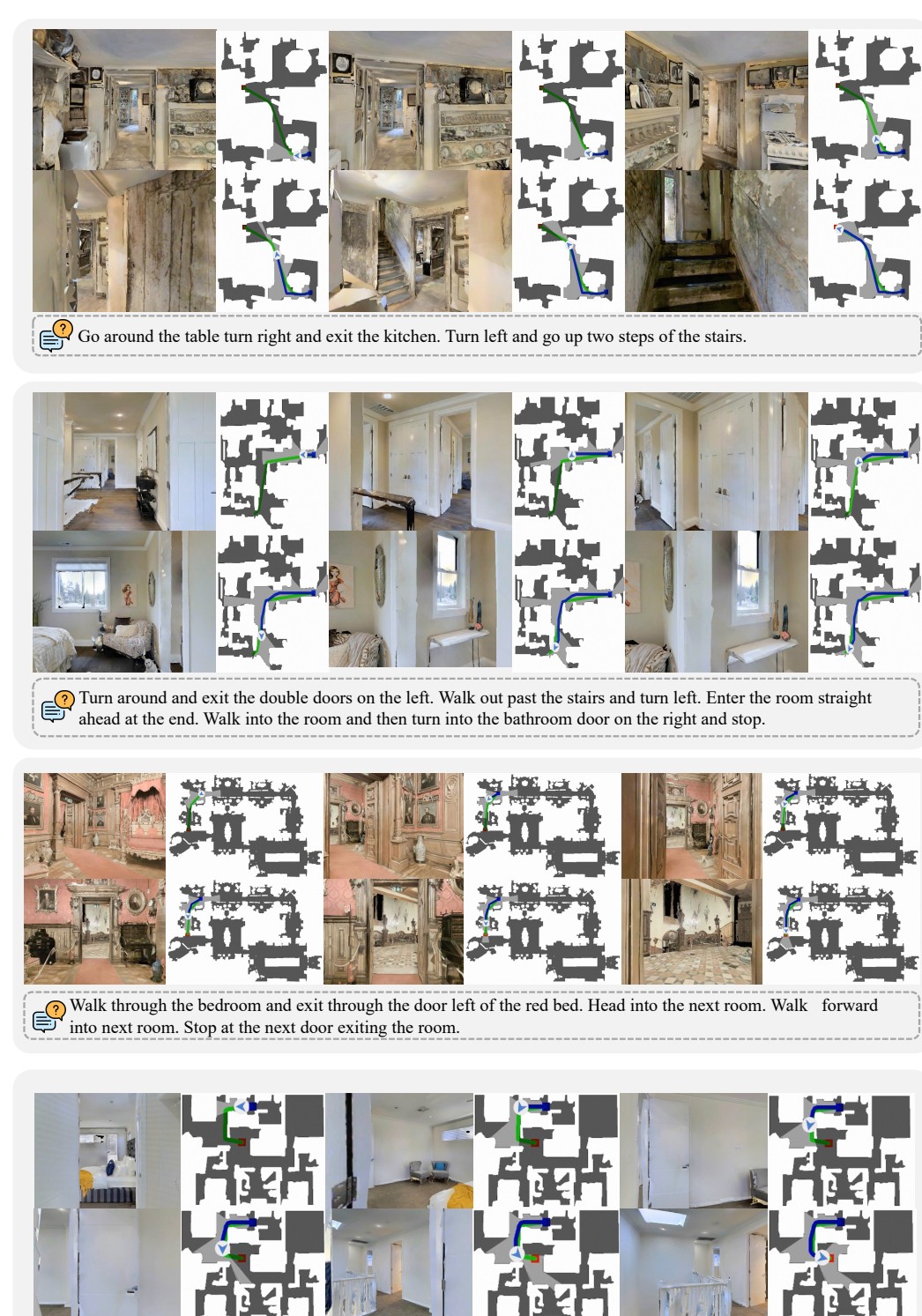

Figure 12: Qualitative results of JanusVLN on R2R-CE.

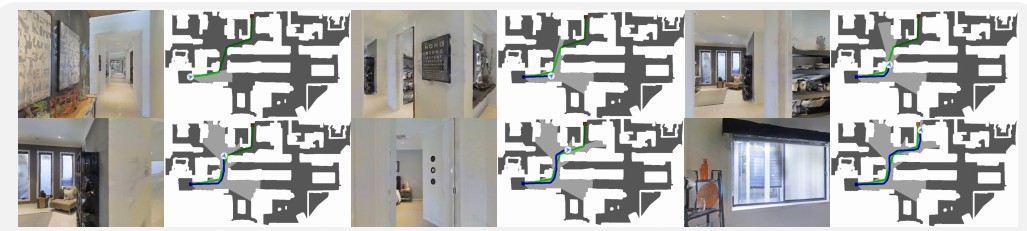

You are standing in a hallway, facing an open, bright area. Go straight down the hallway to an opening on your left, where you'll see a black picture with gold lettering. Take a left. Now you face a room with two windows at the end. Walk a few feet past the first opening, and at the second opening (before the windows), turn right. Continue straight into the bedroom. Once inside, go to the left of the bed, walking between the wall with three pictures and the bed on your right. Walk all the way to the stand with two orange plates. Stand in front of that stand, which has a "Do Not Enter" sign. You are at your destination.

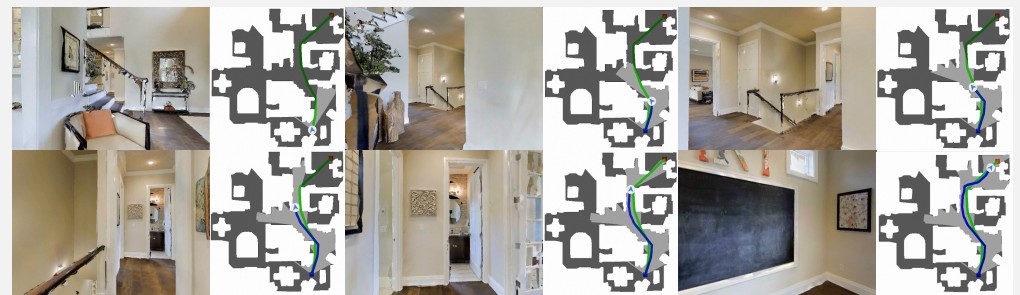

Move forward from the place you are standing, slightly turn to your left and move forward, enter into the area which is to the right of the staircase, you can find an opened door in front of you, move towards the opened door, turn right and enter into the room, turn left, stand in front of the black board, that would be your end point.

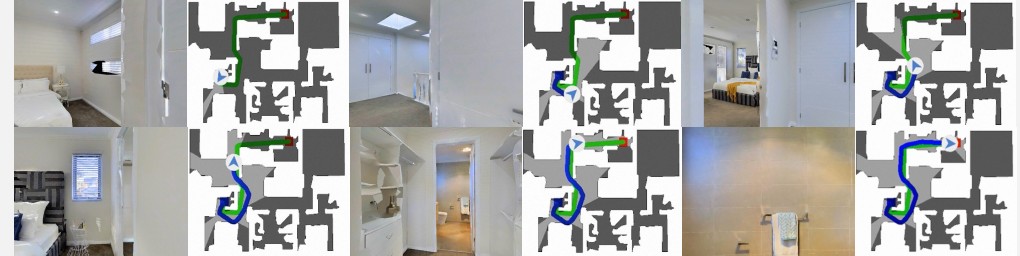

Now you are facing towards the white open door, turn slight left and move towards the bed and take left turn, in front you can see the open door, move towards the door and exit the door, now you are in-front of the white railing, take slight left turn and move forward, you can see the another bed room with an open door, enter into the door take right turn and move towards the bed, in-front you can see the window and white stand, take left "sorry" right turn and move forward and enter into the bathroom, in-front you can see the white towel to the hanger, that is your final destination.



You'll face a pillar and a dining area. Slightly left, find a small wooden table (with a plant and remotes) and couches. Walk along the table towards the kitchen entrance, then one more step towards the wooden cupboard with flowers, and a slight step left. You're now in the kitchen. In the kitchen, to your left are wooden cupboards and a sink, a white flower plant in front, and four glass doors leading outside. Step towards the glass doors, then turn left. You'll see a hallway with four chairs lined up, and a fifth black chair behind them. Step towards that black chair. The glass doors will be on your right, the four chairs on your left, and the fifth chair in front. You are done.

Figure 13: Qualitative results of JanusVLN on RxR-CE.

