# OpenReview forum: "JanusVLN: Decoupling Semantics and Spatiality with Dual Implicit Memory for Vision-Language Navigation"
_ICLR.cc/2026/Conference — ICLR 2026 Poster_

### Official Review · Reviewer_we6n · 2025-10-17

**Soundness:** 3
**Presentation:** 3
**Contribution:** 2
**Rating:** 4
**Confidence:** 4

**Summary:**

The paper introduces JanusVLN, a novel framework for Vision-and-Language Navigation (VLN) that proposes a dual implicit memory system—one for visual semantics and another for spatial geometry. The authors decouple semantic and spatial processing using two separate encoders: a semantic encoder (ViT of Qwen2.5VL) and a pretrained geometric encoder (VGGT). The model use cached key-value (KV) memory from both encoders, updated incrementally via an initial and sliding window strategy, to avoid recomputation and reduce inference overhead. JanusVLN achieves state-of-the-art performance on VLN-CE benchmarks using RGB input alone, surpassing previous methods in both Success Rate (SR) and Success-weighted Path Length (SPL).

**Strengths:**

- **Writing Clarity and Problem Framing**: The paper is generally well-written, with a clear logical flow and a well-motivated problem statement. The limitations of prior work are articulated effectively.
- **Enhanced Spatial Reasoning via Pretrained 3D Geometry Model**: The use of VGGT for feature extraction, trained on pixel-to-point cloud pairs, enables the agent to infer 3D spatial structure from RGB-only input. The approach enhance the capabilitiy of the model to understand 3D environments, and produces improved peformance (shown in ) compared with naive vision encoder of tranditional MLLMs.
- **Efficient Inference**: The KV caching strategy reduces redundant computation, which is quite crucial to enable real-time deployment on physical robots.
- **Strong Empirical Results**: JanusVLN achieves SOTA performance across multiple VLN-CE benchmarks, even outperforming methods that use depth, panoramic views, or odometry.
- **Good Ablation Studies**: The authors provide detailed experiments isolating the impact of spatial memory, semantic memory, and fusion strategies, as well hyper-parameter configurations.

**Weaknesses:**

1.  **KV Cache Contribution Overstated**: While the paper presents KV caching as a core innovation, this technique has been widely used in LLM inference and has been applied in previous VLN approach (e.g., StreamVLN, which also uses KV cache and sliding window for acceleration). The novelty lies more in its integration than in the mechanism itself.
2.  **Architectural Clarity**: Several components lack sufficient explanation:

    - The **attention fusion module** is under-described, and Figure 2 does not clearly illustrate its structure.
    - It is unclear which modules’ KVs are cached—semantic encoder, spatial encoder, or fusion layers?  Specific concerns include:
        - There are no autoregressive attention between frames in the Qwen2.5-VL’s ViT processing process , so caching may not yield meaningful savings.
        - VGGT uses non-causal attention, which typically does not support KV caching. If VGGT is frozen during training, how is its computation reduced during inference?
        - If KV caching is applied only to the fusion module, and that module is lightweight, the overall computational savings may be marginal—especially considering that VGGT is the dominant cost.

3.  **The analysis on the effectiveness of gemetric tokens are insufficient**: The paper does not provide visualization or interpretability analysis of the geometric tokens. This limits understanding of how spatial priors contribute to reasoning.

4.  **Presentation Gaps**: Some figures and descriptions (e.g., memory update mechanism, fusion strategy) could be more precise and better annotated.

**Questions:**

1. Detailed mechnisim of the approach: Please refer to Weakness 2 for concerns regarding the architectural clarity and the caching mechanism. A more thorough explanation of the model components and their interactions is necessary to fully assess the technical soundness of the proposed method.
2. Can the authors provide visualizations or interpretability analysis of geometric tokens to demonstrate their impact on spatial reasoning? It would be beneficial to include qualitative examples or attention maps that illustrate how these tokens contribute to decision-making during navigation, especially in complex spatial scenarios.

---

> ### Author Response · Authors · 2025-11-20
>
> We are sincerely grateful for your insightful comments and meticulous review, which have been instrumental in enhancing the clarity and impact of our work. We have thoroughly revised the manuscript in light of your valuable feedback and offer our itemized responses below:
>
> ---
> > Q1: Regarding KV Cache in the contribution.
>
> A1: We apologize for the misunderstanding and will revise our second core contribution. To clarify, the core contribution of our work is not the KV Cache technique itself, but rather the development of a novel, cognitive science-inspired "Dual Implicit Memory" paradigm. The KV Cache is the tool we selected to implement this implicit memory. Specifically, our innovation is demonstrated on the following two key levels, which fundamentally distinguishes our work from studies such as StreamVLN:
> - Paradigm Innovation: A shift from a "single memory" to a "dual memory" framework. StreamVLN applies the KV Cache to a single stream of visio-semantic features for acceleration. In contrast, our JanusVLN innovatively utilizes the KV Cache to separately construct and maintain two distinct yet complementary types of memory. This decoupled "dual implicit" memory architecture represents a novel exploration in the VLN domain, inspired by the functional lateralization of the human brain's hemispheres.
> - Unlocking Potential: The standard implementation of VGGT reprocesses the entire video history at each step, causing computational costs to grow exponentially with sequence length. As demonstrated in Figure 3 and Table 5, the original VGGT crashes due to OOM errors on a 48G GPU after processing just 48 frames. In contrast, our approach not only implements a KV Cache on VGGT to avoid redundant computations, reducing inference overhead by 69%-90%, but also resolves the GPU memory explosion problem of a naive KV Cache. We achieve this through a dual-window mechanism that uses incremental updates and attention fusion to focus on both the initial frames (as global navigation anchors) and the latest local context. Without our decoupling mechanism and the incremental update and fusion of implicit memory, the direct application of VGGT is extremely inefficient, if not entirely infeasible.
>
> In summary, our contribution is a novel and effective dual implicit memory paradigm for VLN. This paradigm fully unleashes the potential of 3D spatial-geometric foundation models like VGGT in the VLN domain, with the KV Cache serving as the technical enabler of this framework.
>
> ---
> > Q2: Regarding the clarity of architectural.
>
> A2: We apologize for any confusion caused and offer the following detailed clarification:
> 1. Attention Fusion Module:
> - In the original VGGT, frame attention  and global cross-frame attention  are executed alternately. Our spatial encoder, in contrast, fuses information through interaction with a cache during the global attention process. Specifically, the tokens of the current frame first pass through frame attention to establish a local context. Then, during global attention, these current-frame tokens generate the Query. The final Key and Value are constructed by concatenating the historical KV cache with the newly generated KV from the current frame, which are then used to compute the attention. This alternating execution of frame attention and global attention is repeated. **We have added a cyan-highlighted Figure 5 to the Appendix of the revised manuscript to provide a more comprehensive explanation.**
> - Qwen2.5-VL employs the standard KV Cache mechanism typical of LLMs. Visual embeddings derived from new frame via the semantic encoder generate Queries within the language model. These Queries then compute attention against the Keys and Values of all historical tokens combined with the Keys and Values generated by the tokens of the current frame.
>
> 2.KV Cache:
> - For Qwen2.5-VL, we adopt the standard LLM KV caching strategy within the language model, rather than in the visual encoder or fusion module. We cache Key/Value states for prompts and historical image tokens to pre-fill context when new frames arrive, avoiding redundant computation.
> - For VGGT, we enable KV caching by converting global cross-frame attention into causal attention via a masking mechanism that shields future information. This optimization reduces inference time from 1549 ms to 195 ms, significantly lowering costs.
> ---
> > Q3: The analysis on the effectiveness of spatial gemetric tokens
>
> A3: Thank you for your suggestion. In addition to Figure 1, **we have added a cyan-highlighted paragraph "Visualization analysis of spatial geometric tokens" and Figure 10 to the appendix for a more detailed visualization and analysis.** For specific details, please refer to the relevant content in the updated paper.
>
> ---
> > Q4: Presentation gaps.
>
> A4: Thank you for your suggestion. **We have added a cyan-highlighted 'Model structure details' section and Figure 5 to the appendix of the revised manuscript to provide a more comprehensive explanation.**

---

> > ### Comment · Reviewer_we6n · 2025-11-28
> >
> > I appreciate the authors' thoughtful response, which has effectively clarified several key points, most notably regarding the "Dual Implicit Memory" paradigm and the technical details of the KV cache. These clarifications have certainly enhanced the paper's quality.
> >
> > My primary concern has been mitigated. However, regarding the analysis of geometric tokens (Appendix E), the current discussion, while a positive addition, remains primarily heuristic. To strengthen this section for the final version, it would be beneficial to include a more in-depth analysis. For instance, examining the attention weights linked to these tokens could provide more compelling, direct evidence of their role in the model's decision-making process.
> >
> > Overall, the rebuttal has addressed most of my concerns. While the core contribution is incremental, the work is sound and well-motivated. I am therefore leaning towards a weak acceptance​ of the paper for the proceedings.

---

> > > ### Author Response · Authors · 2025-11-28
> > > **Thank you for raising score to the level of weak acceptance！**
> > >
> > > We sincerely thank the reviewer for their valuable time and constructive feedback.
> > >
> > > We are delighted that our response has mitigated the reviewer's "primary concern" and that they now find our work "sound and well-motivated," upgrading their recommendation to "leaning towards a **weak acceptance**." This is highly encouraging.
> > >
> > > We fully agree with the suggestion to strengthen the analysis of geometric tokens. We will incorporate an in-depth analysis of their corresponding attention weights in the camera-ready version to provide more direct evidence of their role.
> > >
> > > Thank you again for the positive feedback. We are committed to diligently implementing these revisions to improve the final paper.

---

### Official Review · Reviewer_UnTs · 2025-10-27

**Soundness:** 2
**Presentation:** 4
**Contribution:** 2
**Rating:** 4
**Confidence:** 4

**Summary:**

This paper introduces a new framework for VLN designed to address the inefficiency of methods that rely on explicit memory.

The authors propose a dual implicit neural memory inspired by the division of labor in the human brain for semantic and spatial processing. This framework decouples visual semantics from spatial geometry, representing each as a compact, fixed-size memory. Technically, this is achieved by caching the key-value (KV) pairs from two separate encoders: a semantic visual encoder from a Multimodal Large Language Model (MLLM) and a 3D visual geometry encoder (VGGT) that infers spatial priors from RGB input alone.

This implicit memory is updated efficiently using an initial window + sliding window strategy, which avoids recomputing features for the entire history at each step. The framework reaches SOTA.

**Strengths:**

(1) I think the way JanusVLN incorporates 3D geometric priors from a model like VGGT, using only RGB video as input, is a major strength. It enhances the agent's spatial awareness without requiring specialized hardware like depth sensors. The community is waiting a better engineering solution that applies VGGT to VLN tasks.

(2) The method achieves SOTA results on challenging benchmarks, including those methods that use additional data types or more training data.

(3) The authors have done an excellent job of dissecting their model and validating each component. The ablations on the dual memory, 3D priors, and memory size are all well-designed and provide strong support for the final architecture.

**Weaknesses:**

My main concern is about the paper's claim on its title: Decoupling Semantics and Spatiality.  Of course the paper indeed successfully argues that both semantic and spatial information are important (as shown by the ablation in Table 3 where removing either hurts performance) . However, it fails to provide a compelling argument for **why** these two types of information must be or had better be decoupled into separate processing streams.

The paper repeatedly draws an analogy to the human brain's hemispheric specialization. While this is an elegant and intuitive starting point, it cannot substitute for a rigorous technical argument. The paper just provides an experimental result which is certainly great. But for the VLN community nowadays, SOTA is indeed good but it is not that improtance or convincing to this specific question. To the best of my knowledge, there is few evidence or analysis to suggest that a standard, unified Transformer architecture would struggle to learn and represent both semantic and spatial features concurrently, or that their deeper fusion would be detrimental. This makes the central claim that **decoupling is necessary or vital** feel more like an assertion than a proven technical need.

**Questions:**

1.My main conceptual question is about the necessity of decoupling. The paper shows that both semantic and spatial information are important. However, could the authors provide a more rigorous technical justification for why these information streams must be processed by separate encoders and fused late, rather than being learned jointly within a single, deeper architecture? Is there any evidence to suggest that a standard Transformer struggles to learn the complex interplay between semantics and spatial geometry, making this explicit separation necessary? I think a clear answer (either theoretical or experimental) to this would significantly strengthen the paper's conceptual contribution, perhaps even more so than the SOTA results.

2.Following on from the fusion mechanism, I'd be very interested to know the authors' thoughts on why the simple addition fusion outperformed Cross-Attention. Do you have any hypotheses that go beyond what's presented? Could it be that the two feature streams are already well-aligned by the powerful encoders, making complex interactions unnecessary or even noisy?

3.I'm curious about the generalizability of the spatial priors. VGGT is pre-trained on pixel-3D point cloud pairs. I wonder how robust the system would be if deployed in environments with architectural or visual styles that differ significantly from VGGT's pre-training data. Have you considered this potential domain gap?

These main points are based on the improvement and insights that can benefit the community. If I do have any misunderstanding welcome to point me out.

---

> ### Author Response · Authors · 2025-11-20
>
> Thank you for your thoughtful review and constructive feedback, which have been invaluable in helping us refine our work. We appreciate the opportunity to clarify our design choices and experimental findings:
>
> ---
>
> > Q1: Regarding decoupling semantics and spatiality.
>
> A1: Thank you for your insightful and constructive questions. First, we would like to make the following clarifications:
>
> - In the field of VLN, existing work typically overlooks the 3D spatial information implicit in 2D images, and there is no coupled model to process semantic and spatial information simultaneously. Our work, JanusVLN, represents a key step forward by attempting to extract spatial information solely from 2D images through a decoupling approach.
>
> - Currently, there is no conclusive evidence to suggest that a unified Transformer architecture is inherently unable to learn and represent both semantic and spatial features simultaneously. However, no existing unified Transformer has demonstrated the ability to do so with high efficacy. These models typically exhibit stronger semantic capabilities than spatial understanding, or they require additional 3D data as input to bridge the gap. Developing a coupled model that handles both semantic and spatial information within a single framework is a valuable direction for future work.
>
> Next, we will explain from technical and experimental perspectives why we currently opt for a decoupled and complementary approach:
>
> - Technical Rationale: Complementary Pre-trained Tasks and Inductive Biases
>   - Our choice for a decoupled architecture stems not from the inability of a unified model, but from leveraging the respective strengths of distinct 'expert models'. Our core motivation is that instead of training a large-scale, unified Transformer from scratch to simultaneously learn two disparate types of representations (2D semantics vs. 3D geometry), it is more effective and efficient to integrate two pre-trained, SOTA "expert" models. These expert models possess complementary inductive biases. Our decoupled and complementary architecture preserves and maximizes the respective unique strengths of these two expert models.
>
> - Direct Experimental Evidence from Table 4
>
>   - The marginal performance gain from adding another powerful encoder (e.g., DINOv2, SigLIP 2) to our baseline demonstrates that simply stacking more homogenized semantic information is ineffective for addressing the bottleneck in spatial understanding. This directly answers your question about whether a deeper architecture would work: it offers little benefit if the type of information remains unchanged. In contrast, our decoupled approach, which introduces an expert model with different inductive biases, provides an effective and efficient solution. While a unified framework could potentially be built through a new paradigm that integrates 2D semantic and 3D spatial pre-training, this represents a substantial yet meaningful undertaking that we leave for future exploration.
>
>
> ---
>
> > Q2: The reason why weighted addition is superior to cross-attention.
>
> A2: While we concur with the hypothesis that a powerful encoder might have already well-aligned the feature streams, the following reasons still apply:
>
> - Model Simplicity and Robustness: The fusion module must learn from limited downstream data. Our weighted-addition fusion (with a lightweight MLP) is parameter-efficient and structurally simple, making it easier to train and less prone to overfitting. This leads to enhanced robustness and generalization. Conversely, the more complex and parameter-heavy Cross-Attention module is harder to optimize sufficiently with limited data, making it susceptible to overfitting specific patterns and resulting in suboptimal performance on unseen scenes.
>
> - Inherent Spatial Alignment: Our semantic and spatial feature maps are inherently aligned, as they originate from the same image and share identical spatial resolution. This means that at any given location, a semantic token and a spatial token describe the semantic and geometric attributes, respectively, of the same physical region. Weighted addition directly injects the MLP-projected geometric information into the semantic features at corresponding locations, precisely enhancing the spatial awareness of each local feature while perfectly preserving this valuable spatial correspondence.
>
> - Future Work: Finally, exploring more sophisticated and effective fusion methods is a promising direction for future research.
>
>
> ---
>
> > Q3: The generalizability of the spatial priors.
>
> A3: We acknowledge the issue of VGGT's performance variation across different domains. However, VGGT is designed as a general-domain 3D foundation model, trained on a large and diverse dataset. This dataset covers various domains, including indoor/outdoor environments and both simulated/real-world scenes, which closely aligns with the domain of our experimental data, showing no significant discrepancy.

---

### Official Review · Reviewer_y4hH · 2025-10-31

**Soundness:** 4
**Presentation:** 3
**Contribution:** 3
**Rating:** 6
**Confidence:** 4

**Summary:**

This paper introduces a novel semantic-spatial dual-memory mechanism for visual language navigation tasks. The semantic and spatial memories are stored as KV caches extracted from VLM visual encoder and VGGT representation, respectively. It demonstrates superior performance compared to SOTA baselines in R2R/RxR-CE datasets, with lower computational overhead. Overall the proposed method is novel and effective. The experimental setup and research methodology are sound. My main concern is the lack of open-sourced implementation of the method, which impedes full verification of the results and reproducibility.

**Strengths:**

1. This work addresses a critical task that remains highly challenging for current embodied systems. Furthermore, the selection of baselines is comprehensive, incorporating all recent state-of-the-art advancements.

2. The proposed semantic-spatial dual-memory mechanism is both novel and highly effective, demonstrating superior performance over all baselines on both the R2R and RxR-CE datasets.

3. The proposed method has been successfully deployed on a real robot platform. Qualitative results are provided.

4. The ablation studies are thorough and complete, evaluating the impact of critical design choices, including geometric prior selection and memory size.

**Weaknesses:**

1. The lack of open source code for the proposed method impedes full verification of its reproducibility.

**Questions:**

1. NaVILA and StreamVLN are the two strongest baselines shown in Table 1 and 2. However, they are both trained with different datasets. Comparison to these two approaches trained on the same 10.6k extra data used by JanusVLN will rule out the impact of training data. Could you run such experiments?

2. Any plan to open-source the proposed method?

---

> ### Author Response · Authors · 2025-11-20
>
> We offer our sincere thanks for the time and effort you have dedicated to reviewing our manuscript. We have carefully engaged with each of your insightful comments, and our detailed responses are provided below:
>
> ---
>
> > Q1: Open source.
>
> A1: We fully agree with the reviewer on the importance of reproducibility and community contribution. We are pleased to report that we have fully open-sourced the code, model weights, and data for JanusVLN. We are encouraged to see that it has already garnered widespread attention and positive feedback from the VLN community. We hope this work facilitates other researchers in building upon it and collectively advancing the field of VLN.
>
>
>
> ---
>
> > Q2: Regarding the use of different datasets for training.
>
> A2: Thank you again for raising this insightful question. You have astutely pointed out that the discrepancy in training data is a significant challenge for model comparison in the current Vision-and-Language Navigation (VLN) landscape.
>
> We would like to take this opportunity to clarify and elaborate on the specifics of JanusVLN:
>
> - Our Data Composition: As shown in Tables 1 and 2 of our paper, the training data for JanusVLN (10.6M) is composed of widely recognized benchmark datasets within the field, including R2R-CE, RxR-CE (excluding EnvDrop), ScaleVLN, and the DAgger dataset.
>
> - Comparative Analysis with SOTA Methods: When conducting our comparisons, we indeed noted that other state-of-the-art methods, such as StreamVLN and NaVILA*, incorporate additional and varied datasets beyond the standard ones. This makes a perfectly controlled, fair comparison exceptionally challenging. However, it is precisely in this context that we observed a phenomenon that further highlights the advantages of JanusVLN:
>
>   - Versus StreamVLN: Despite StreamVLN integrating a large volume of additional data on top of the datasets we used, JanusVLN still outperforms it by a significant margin, with a 3.6 point lead in SR and a 4.9 point lead in SPL.
>   - Versus NaVILA*: Similarly, while NaVILA* shares the R2R-CE and RxR-CE datasets with JanusVLN*, it also utilizes a substantial amount of extra data. Even so, JanusVLN* leads with an SR of 52.8 (vs. 49.7) and an SPL of 49.2 (vs. 45.5), representing advantages of 3.1 and 3.7 points, respectively.
>
> - Our Conclusion: As you have insightfully observed, perfectly aligning the supplementary training data across different methods is difficult, and conducting an "ideal experiment" with strictly controlled conditions presents objective challenges in the current research ecosystem. Therefore, we believe the fact that JanusVLN achieves superior performance while being at a relative "disadvantage" in terms of training data scale serves as stronger evidence for the efficiency and excellence of our proposed method. This indicates that our model architecture is capable of learning more effectively from data, rather than simply relying on data accumulation.

---

> > ### Comment · Reviewer_y4hH · 2025-11-24
> >
> > Thanks for the response. I just saw the open-sourced repository. Overall I believe the work is solid and I would like to keep my rating.

---

> > > ### Author Response · Authors · 2025-11-25
> > >
> > > Thank you very much for your timely response and for taking the extra time to review our open-sourced repository. We are greatly encouraged by your assessment that our work is "solid." Your valuable feedback throughout the review process has been very helpful.

---

### Official Review · Reviewer_KxJm · 2025-11-02

**Soundness:** 3
**Presentation:** 3
**Contribution:** 3
**Rating:** 6
**Confidence:** 4

**Summary:**

The paper proposes a novel approach for Vision-and-Language Navigation (VLN) by introducing a framework that effectively decouples semantic and spatial-ity information using a Dual Implicit Memory mechanism. The paper presents a logically sound architecture that aims to overcome the limitations of explicit semantic memory in VLN agents. The empirical results demonstrate that the proposed method significantly advances the state-of-the-art performance across several established VLN benchmarks.

**Strengths:**

1. The paper is well-structured and clearly written. The core ideas, particularly the decoupling mechanism and the design of the dual implicit memory, are logically presented, making the methodology straightforward to understand.

2. The empirical results are highly compelling. The proposed method achieves excellent performance and reports SOTA results on the evaluated, well-known VLN benchmarks. This suggests a notable improvement in the agent's navigation and generalization capabilities.

**Weaknesses:**

1. The primary performance evaluation heavily relies on relatively older and more indoor/limited benchmarks. While the paper includes some results on real-world scenarios, these are unfortunately only qualitative results, completely lacking quantitative metrics. The submission would be significantly stronger if it included quantitative validation on real-world data and performance on more recent, challenging, and diverse VLN benchmarks from the past 3 years.

2. A critical concern is that the proposed method appears to be highly dependent on the capability of the underlying VGGT model. This dependency raises questions about the independent contribution of the novel framework components versus the inherent power of the advanced backbone itself. This requires clearer justification and comparative analysis.

3. The experimental analysis of the overall framework's effectiveness and validity is not deep enough. The ablation studies provided are basic. There is a lack of detailed analysis on: (a) the specific scenarios where the proposed method successfully outperforms others, and (b) a systematic investigation of the current failure modes of the agent. A comprehensive analysis of successful vs. failing cases is crucial for the community to understand the method's limits and aid future improvements.

**Questions:**

1. Given the goal of generalizable navigation, the lack of quantitative real-world validation is a significant drawback. Could the authors provide quantitative results for the real-world scenarios presented? Additionally, please elaborate on the practical and operational details of the real-world experiments, such as sensor configuration, map representations used (if any), and any specific data preprocessing required for deployment.

2. Regarding the dependency on the VGGT model (Weakness 2), please offer a more extensive explanation and stronger comparative evidence to clarify the unique contribution of the decoupling mechanism and the Dual Implicit Memory structure. Specifically, how does this memory module provide a benefit that a simpler, standard memory module would not, when both are paired with the exact same VGGT backbone?

3. To improve the analytical depth (Weakness 3), please strengthen the experimental validity analysis. This should include providing more detailed examples and, ideally, a systematic statistical analysis of the agent's behavior (e.g., analysis of error types, performance on various instruction lengths/complexity, or specific semantic grounding success rates) to better guide and inspire related future research.

---

> ### Author Response · Authors · 2025-11-20
>
> Thank you for your valuable time and insightful comments. We have addressed your comments point-by-point as follows:
>
> ---
> > Q1: Quantitative results from real-world and other recent datasets.
>
> A1:
> 1. We fully agree with your insight that quantitative real-world evaluation is crucial. While establishing a perfectly fair comparison is challenging due to the lack of unified protocols (e.g., differences in initial pose precision and dynamic environments), we have conducted a rigorous quantitative ablation study specifically to address your concern. **The details are presented in the "Real world quantitative results" paragraph (highlighted in yellow) and Figure 6 in the appendix of the updated paper.**
>
> | Method | Total | Normal Navigation | Spatial Understanding |
> | :--- | :--- | :--- | :--- |
> | JanusVLN(w/o Spatiality) | 58.6 | 64.4 | 53 |
> | JanusVLN | 73.3 | 71.1 | 76.6 |
>
> 2. For quantitative real-world evaluation, we used 25 instructions, each repeated three times. A trial is considered successful if the robot stops within 1 meter of the target. In the table above, JanusVLN outperforms its variant without spatial memory across all scenarios. Notably, it achieves a 23.6% improvement on spatial understanding navigation tasks, which demonstrates the effectiveness of JanusVLN.
> 3. We primarily focused on VLN tasks, using the representative VLN-CE benchmark. Following your suggestion, we have also evaluated our method on the more diverse and recent HM3D-OVON benchmark on the below table, which emphasizes Object-Goal Navigation. Our method boosts the SR from 40.8% to 44.9% and demonstrating excellent generalization.
>
> | Method | SR | SPL |
> |:---|:---|:---|
> | VLFM [ICRA24] | 35.2 | 19.6 |
> | DAgRL+OD [IROS24] | 37.1 | 19.8 |
> | Uni-NaVid [RSS 25] | 39.5 | 19.8 |
> | MTU3D [ICCV 25] | 40.8 | 12.1 |
> | JanusVLN | 44.9 | 31.7 |
> ---
> > Q2: Regarding the dependency on the VGGT model.
>
> A2: Thank you for this crucial question. We appreciate the opportunity to clarify that our decoupled framework is not dependent on any specific encoder; rather, it proposes a general dual implicit memory paradigm to enhance navigation capabilities by fusing complementary 2D semantic and 3D spatial information. For the first time, it addresses the challenge of how to efficiently and feasibly leverage such powerful 3D priors in a streaming navigation task.
>
> Contribution 1: Decoupling semantics from spatial and verifying their complementarity, rather than merely adding another encoder:
> - Evidence of Complementarity (Table 4): Replacing VGGT with other powerful 2D vision encoders (e.g., DINOv2, SigLIP 2) yields very limited performance gains. This is because they provide 2D semantic information that is redundant to the Qwen2.5-VL encoder. In contrast, VGGT, pre-trained on pixel-to-3D-point-cloud pairs, offers complementary 3D spatial geometric priors, resulting in a substantial 5.8 increase in the SR.
> - Evidence for the Necessity of the 3D Prior (Table 4): A randomly initialized VGGT (row 4) brings almost no performance gain. This proves that the improvement stems not from an increased parameter count but from the 3D geometric prior knowledge embodied within the pre-trained VGGT.
> - In summary, our decoupled framework introduces a general dual implicit memory paradigm that enhances navigation capabilities by fusing complementary 2D semantic and 3D spatial information. This approach is not reliant on any specific encoder but rather on the 3D geometric priors it provides. The paradigm can be further strengthened in the future by incorporating more advanced spatial encoders.
>
> Contribution 2: Implicit memory for efficiency and high performance:
> - Advantage 1: Computational Feasibility
> A standard memory approach with VGGT re-processes the entire video history at each step, causing computational overhead to grow exponentially. As shown in Figure 3 and Table 5, the vanilla VGGT fails with OOM after processing just 48 frames. In contrast, our implicit memory reduces inference overhead by 69%-90% by caching and reusing historical KV pairs with dual-window incremental updates.
> - Advantage 2: Superior Performance
> As detailed in Table 5, our implicit memory not only has lower inference overhead but also outperforms a simple historical frame baseline, improving SR by 2.6 and SPL by 4.2. It achieves this by retaining the initial frame as a global navigation anchor while using a sliding window to focus on local context.
> - Therefore, our dual implicit memory module is the key to making VGGT efficient and practical for streaming navigation, an objective unachievable with standard memory approaches.
> ---
> > Q3: In-depth statistical analysis.
>
> A3: Thank you for your suggestion. **We have added "Statistical analysis" section, highlighted in yellow, to the appendix for a more detailed analysis.** This section includes the "Success and strengths analysis" and Figure 7, the "Performance by instruction length" and Figure 8, and the "Failure case analysis" and Figure 9.

---

> > ### Comment · Reviewer_KxJm · 2025-11-27
> >
> > Thank you for your response. Most of my concerns have been addressed, and I will maintain my positive score.

---

> > > ### Author Response · Authors · 2025-11-27
> > >
> > > Thank you very much for your prompt reply and positive feedback. We are glad that our revisions and responses have addressed your main concerns. We sincerely appreciate your valuable time and insightful comments, which have significantly improved the quality of our manuscript.

---

### Author Response · Authors · 2025-11-29
**Summary of Rebuttal Discussion and Consensus**

Dear Area Chair and Reviewers,

We sincerely thank you for your time and invaluable feedback, which has significantly improved the quality of our paper. We are grateful for the constructive discussion and are pleased to see a strong consensus emerging from the post-rebuttal feedback.

Following our response, we are delighted that Reviewers KxJm, y4hH, and we6n have all confirmed that their primary concerns have been successfully addressed.

- Reviewer KxJm confirmed: "Most of my concerns have been addressed, and I will maintain my **positive score**."
- Reviewer y4hH, after reviewing our open-source repository, concluded: "Overall I believe the work is **solid** and I would like to keep my rating."
- Reviewer we6n noted our response "has effectively clarified several key points" and "mitigated primary concern," leading them to **lean towards acceptance**. We appreciate their suggestion for a deeper analysis of geometric tokens and will gladly incorporate this into the updated version to further strengthen the paper.
- While we await a follow-up from Reviewer UnTs, we are confident that our detailed rebuttal addresses their points. We note that their initial review praised several core strengths that align with the consensus from other reviewers.

Across all reviews, a clear and consistent picture of our paper's contributions has emerged. We are encouraged that all reviewers acknowledged the following key strengths:

- **Novel and Effective Architecture**: The proposed Dual Implicit Memory mechanism for decoupling semantic and spatial information was consistently highlighted as novel, effective, and logically sound (praised by KxJm, y4hH, we6n, UnTs).

- **State-of-the-Art Performance**: Our method's compelling empirical results and achievement of SOTA performance on major VLN benchmarks (R2R, RxR-CE) were recognized as a major strength by all reviewers (KxJm, y4hH, we6n, UnTs).

- **Innovative Use of Geometric Priors**: The integration of a pretrained geometric encoder (VGGT) to infer 3D spatial structure from RGB-only input was lauded as a significant technical contribution that enhances spatial awareness without specialized hardware (praised by we6n, UnTs).

- **Thorough and Sound Evaluation**: Our comprehensive ablation studies, extensive baseline comparisons, and clear presentation were all commended, confirming the soundness of our work (praised by KxJm, y4hH, we6n, UnTs).

In summary, the review process has confirmed that our work is sound, well-motivated, and presents a significant advance in Vision-and-Language Navigation. There is a clear consensus among the reviewers regarding the paper's technical novelty, strong empirical validation, and overall quality.

We thank you once again for your guidance and positive consideration. We are confident that our paper and open-source repository is a solid and impactful contribution to the community.

Sincerely,

The Authors

---

### Meta-Review · Area_Chair_FUok · 2026-01-03

**Summary:**

**Overall:** reviewers broadly agree the paper is sound and strong empirically, with the main debate being conceptual novelty/justification (why “decoupling” is necessary) and technical clarity (what exactly is cached, where speedups come from), plus validation breadth (real-world quant + newer benchmarks).

### Commonly recognized strengths
- **Dual implicit memory** framing (separate semantic vs spatial memory) is seen as novel/effective by multiple reviewers.
- **Strong SOTA results** on VLN benchmarks (R2R / RxR-CE etc.), beating many baselines including those with extra modalities/data.
- **Using VGGT 3D priors from RGB-only** is viewed as a meaningful technical contribution for spatial reasoning without depth sensors.
- **Efficiency focus** (avoiding recomputation / memory bloat) is valued, especially for real-time robotics.

### Main concerns raised
- **Benchmark breadth / real-world quantification**: early review wanted quantitative real-world evaluation and newer, more diverse benchmarks (not only “older indoor” ones).
- **Dependency on VGGT**: is the gain mostly “because VGGT is strong,” vs the proposed memory/decoupling being the real contribution?
- **Conceptual necessity of decoupling** (Reviewer UnTs): the paper shows both semantics and spatial info matter, but doesn’t initially prove that *separating* them is better/necessary vs a unified architecture.
- **KV-cache novelty / clarity** (Reviewer we6n): caching is not new; plus confusion about *where caching is applied* (Qwen visual encoder? LLM? VGGT cross-frame attention?) and whether the claimed speedups are plausible given non-causal attention.
- **Interpretability of geometric tokens**: request for more direct evidence (e.g., attention weights) about their role in decisions.
- **Reproducibility**: needed open-source code/weights.

**Reviewer Concerns:**

## Concerns that the rebuttal addressed

**1. Reproducibility / open source (y4hH’s main issue)**

Authors state they **open-sourced code, weights, and data**, and reviewer y4hH confirmed seeing the repo and kept rating.


**2. Real-world quantitative evaluation & newer benchmark (KxJm’s concern)**

Authors added:
- **Quantitative real-world ablation** (25 instructions × 3 repeats; success = stop within 1m), with a “w/o spatiality” variant vs full JanusVLN (73.3 vs 58.6 total; large gain on “spatial understanding” slice).
- **HM3D-OVON** results (SR and SPL), positioning it as a newer, diverse benchmark.
Reviewer KxJm responded: “Most of my concerns have been addressed” and maintained a positive score.



**3. “It’s just VGGT” dependency / isolating contributions**

Authors argue and provide evidence that:
- Simply adding stronger **2D encoders** yields little gain (redundant with Qwen semantics), while VGGT adds complementary 3D priors and boosts SR.
- A **randomly initialized VGGT** gives almost no improvement, suggesting gain is from pre-trained 3D priors not parameter count.
- Dual implicit memory is needed to make VGGT feasible: vanilla VGGT with history reprocessing hits OOM; their cache/update design reduces overhead and improves SR/SPL vs a simpler historical-frame baseline.


**4. KV-cache novelty + architectural clarity (we6n’s critique)**

Authors explicitly reframed the contribution: not “KV cache is novel,” but **dual implicit memory paradigm**, with KV cache as implementation tool.
They also clarified:
- For **Qwen2.5-VL**, caching is in the language model KV (prompts + historical image tokens), not in the visual encoder.
- For **VGGT**, they enable caching by converting cross-frame attention into causal attention via masking, reducing inference time substantially (example numbers given).
Reviewer we6n said primary concern was mitigated and leaned weak accept, asking only for deeper token analysis.


**5. Added deeper analysis (instruction length, failure cases, etc.)**

Authors added an appendix “Statistical analysis” section with:
- success/strength analysis,
- performance vs instruction length,
- failure case analysis.

** 6. Decoupling justification (UnTs)**

Authors’ response is candid:
- No proof a unified Transformer cannot do it.
- Their rationale is leveraging two pretrained expert models with complementary inductive biases, and empirical evidence that “more 2D semantics” doesn’t fix spatial bottleneck while 3D priors do.

---

## 3. Concerns that remain outstanding (fully or partially)

**1. “Necessity” of decoupling is still not rigorously established**

UnTs’ key point isn’t fully “proven away.” The authors mostly reposition the claim:
- decoupling is pragmatic and effective given the current pretrained components,
- not theoretically necessary.

**2. Geometric token analysis is still somewhat heuristic**

Reviewer we6n explicitly says the added analysis is a “positive addition” but remains “primarily heuristic,” requesting attention-weight analysis tied to those tokens.

Authors agree to add this in camera-ready.


**3.  Training-data controlled comparisons weren’t performed**
Reviewer y4hH asked for NaVILA/StreamVLN trained on the same extra data; authors explain why this is hard and argue JanusVLN wins despite being “data-disadvantaged.”

**Reviewer Scores:**

All reviewers seem to keep or increase their scores to or above 6.

---

### Decision · Program_Chairs · 2026-01-26

Accept (Poster)